# Multi-biosensing hairband for emergency health assessment

Ming Li[1], Ganghua Li[1], Tong Xu[1], Yiwen Wang[1], Ruidong Xu[1], Xinwei Zhang[1], Fuxing Chen[1], Ning Yu [2] ✉ & Mingwei Tian [1] ✉

Blood analysis is regarded as the gold standard for monitoring analytes in clinical diagnostics. However, the time-consuming and site-limited nature often delays medical interventions that are crucial for patients in emergency scenarios. Herein, a weavable multi-biosensor array is developed through coaxial wet spinning for real-time multiplex detection of sweat biomarkers (pH, $Na^+$, $K^+$, and $Ca^{2+}$) and body temperature. The engineered microstructure of the multi-biosensor array exhibits a surface area that is around 200 times larger than that of conventional coated yarns, thereby facilitating directional sweat transport. The sensitivities of the pH, $Na^+$, $K^+$, $Ca^{2+}$ and temperature sensors have been determined to be $39.52 \pm 0.5$ mV $pH^{-1}$ (3-7), $56.33 \pm 1$ mV $dec^{-1}$ (10-160 mM), $34.13 \pm 0.6$ mV $dec^{-1}$ (2-32 mM), $30.61 \pm 0.8$ mV $dec^{-1}$ (0.5-2.53 mM), and $1.2\ \Omega \pm 0.02\ °C^{-1}$ (25-45 °C), respectively. It is noteworthy that the multi-biosensors exhibit consistent operational stability over 24 hours with minimal signal drift (pH: $0.13 \pm 0.01$ mV $h^{-1}$, $Na^+$: $0.17 \pm 0.02$ mV $h^{-1}$, $K^+$: $0.1 \pm 0.008$ mV $h^{-1}$, $Ca^{2+}$: $0.19 \pm 0.01$ mV $h^{-1}$, temperature: $0.05 \pm 0.004\ \Omega\ h^{-1}$). By integrating the multi-biosensors and a circuit patch into the textile substrate, a wireless hairband system is constructed for tracking human physiological dynamics. Such significant technological advancement offers an innovative strategy for constructing real-time biosensing systems, which have the potential to revolutionize personalized healthcare and enable early diagnosis in emergency situations.

Blood analysis remains an indispensable tool in medical monitoring, providing critical physiological and pathological information to support disease diagnosis, therapeutic evaluation, and dynamic health monitoring[1,2]. However, the reliance on invasive sampling and centralized laboratory infrastructure restricts real-time monitoring capabilities in emergencies requiring immediate clinical intervention. To overcome these constraints, extensive research has focused on developing real-time, autonomous biomarker analysis systems utilizing non-invasive biofluids[3,4]. Among alternative biofluids, sweat has emerged as a promising medium due to the ubiquitous distribution of sweat glands across the human body[5,6]. In addition, sweat is rich in biomarkers,

including electrolytes, metabolites, nutrition, and hormones, which reflect physiological conditions at the molecular level and track health status[7–9]. Recent studies have demonstrated clinical correlations between sweat biomarkers concentrations and corresponding blood analyte levels[10–12], providing a scientific foundation for sweat-based diagnostics. Although sweat offers unique advantages for real-time health monitoring, significant challenges remain in practical implementation. Firstly, dynamic compositional fluctuations and ultralow biomarker concentrations necessitate exceptional sensitivity and ultra-low detection limit[13,14]. Secondly, reliance on passive secretion or external stimulation limits continuous acquisition of samples[15,16]. These

---

[1]Research Center of Health and Protective Smart Textiles, State Key Laboratory of Bio-Fibers and Eco-Textiles, College of Textiles and Clothing, Qingdao University, Qingdao, P.R. China. [2]Department of Anesthesiology, The Affiliated Hospital of Qingdao University, Qingdao, P.R. China. ✉e-mail: yuning@qduhospital.cn; mwtian@qdu.edu.cn

intrinsic constraints have motivated continuous advancements in sweat analysis system for active biofluid harvesting and precise biomarker quantification.

Electrochemical biosensors dominate sweat analysis owing to their high sensitivity, selectivity, and miniaturization potential[17,18]. Based on transduction mechanisms, electrochemical detection can be categorized into six modalities[19]: potentiometric, amperometric, voltammetric, organic electrochemical transistors (OECTs), photo-electrochemical (PEC), and electrochemiluminescence (ECL). Potentiometric configurations are prevalent for ion-selective detection due to operational simplicity, while amperometric approaches excel in metabolite quantification through redox reactions. As an early innovation, Gao et al.[20] pioneered the fully integrated wearable biosensor array on poly(ethylene terephthalate) (PET), combining potentiometric and amperometric modalities for multiplexed sweat analysis. Subsequent developments have also employed flexible polymeric substrates, such as polydimethylsiloxane (PDMS)[21–23] and polyimide (PI)[24,25], to enhance mechanical compliance with epidermal surfaces. However, the inherent hydrophobicity of polymer impedes interfacial fluid transport, thereby undermining continuous biosensing reliability. While microfluidic integration enhances biofluid collection, intricate fabrication and high cost remain unresolved issues. Furthermore, poor breathability often induces skin irritation during prolonged wear, presenting critical barriers to the long-term application of wearable biosensors.

Textiles with hierarchical fiber architectures spanning micro-to-nano scales provide distinct advantages for wearable integration, in terms of breathability, lightweight, and intrinsic compatibility with garment systems[26]. While conventional textile biosensor fabrication relies on drop-coating or printing techniques for functional layer deposition, these methods often exhibit structural inhomogeneity and performance variability[27]. The deficiencies are attributable to weak interfacial binding energy between textile substrate and functional layer, resulting in progressive delamination and performance degradation[28]. Such interfacial failures represent fundamental limitations of additive manufacturing strategies based on post-fabrication functionalization. In contrast, wet spinning is a monolithic yarn formation technique, whereby spinning dope is extruded into a coagulation bath and further utilized solvent-nonsolvent dual diffusion to induce phase separation[29–31]. Incorporating biosensing elements into spinning dope enables the fabrication of monolithic sensing yarns with seamless conductive-sensing networks, while also achieving shortened electron transfer pathways and directional sweat transport[32,33]. Furthermore, the comfortable yarn structure mitigates allergic or inflammatory reactions during direct skin contact with textile-based biosensors[34,35]. Such systems mark a significant advancement in wearable diagnostics by eliminating interfacial delamination risks, thereby establishing a robust platform for emergency healthcare monitoring through reliable wearable sensor systems.

Herein, we report a weavable multi-biosensing hairband designed for wireless monitoring in emergency health assessments. The integrated hairband system incorporates an electrochemical multi-biosensor array for sweat biomarker detection, a temperature sensor for human body temperature monitoring, and customized circuitry for signal recording, processing and wireless transmission. The multi-modal biosensors, fabricated through coaxial wet spinning technology, achieve high sensitivities of $39.52 \pm 0.5\,\mathrm{mV\,pH^{-1}}$ (pH), $56.33 \pm 1\,\mathrm{mV\,dec^{-1}}$ (Na$^+$), $34.13 \pm 0.6\,\mathrm{mV\,dec^{-1}}$ (K$^+$), $30.61 \pm 0.8\,\mathrm{mV\,dec^{-1}}$ (Ca$^{2+}$), and 1.2

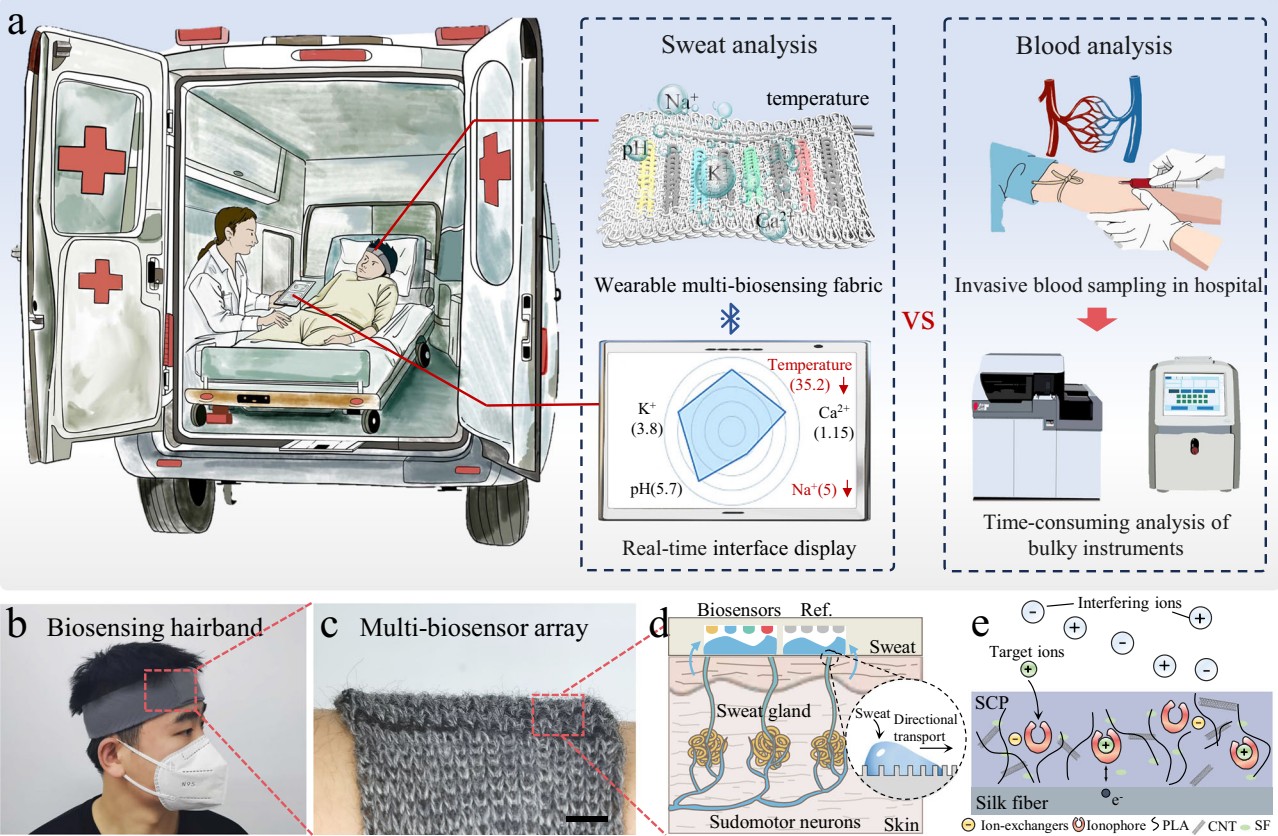

**Fig. 1 | The multi-biosensing hairband system for emergency health assessment. a** Illustration of the multi-biosensing hairband system that continuously monitors physiological signals from sweat in emergency situations with a comparison to blood analysis. **b, c** Optical image of a multi-biosensing hairband system put on a subject (**b**) and the multi-biosensor array within the hairband system (**c**). Scale bar, 0.5 cm. **d** Mechanism of the multi-biosensing hairband for directional sweat capture. **e** Sensing mechanism of as-spun biosensors in sweat. SCP, SF/CNT/PLA mixture; CNT, carboxylated carbon nanotubes; PLA, polylactic acid; SF, silk fibroin.

$\Omega \pm 0.02\ \Omega\ °C^{-1}$ (temperature), respectively. Furthermore, the as-spun biosensors ensure stable performance for up to 24 hours, with drifts of $0.13 \pm 0.01$, $0.17 \pm 0.02$, $0.1 \pm 0.008$, $0.19 \pm 0.01\ mV\ h^{-1}$ for pH, $Na^+$, $K^+$, $Ca^{2+}$ sensing, and $0.05 \pm 0.004\ \Omega\ h^{-1}$ for temperature sensing. Due to the eco-friendly materials and comfortable structural design, the favorable biodegradability and biocompatibility of the biosensors enable prolonged direct contact with human skin without irritation. Furthermore, the as-spun biosensors exhibit uniform porous structures that enhance electron transfer and promote directional sweat transport. The accuracy and reliability of biosensors are validated under various conditions (e.g., deformation, washing, and temperature fluctuations), which ensures real-time monitoring of sweat composition and body temperature. This innovative approach to constructing the multi-biosensing hairband system provides valuable insights for developing advanced wearable health monitoring platforms capable of comprehensive and accurate assessment of human health status.

## Results

### Design and construction of multi-biosensing hairband

In order to provide the real-time physiological information, a multi-biosensing hairband system with high sensitivity and rapid response has been developed for emergency health assessment (Fig. 1a and Supplementary Fig. 1). Unlike the blood analysis, which is time-consuming and necessitates the use of bulky laboratory apparatus, the multi-biosensing hairband is portable and enables real-time tracking in a non-invasive way, offering as a valuable tool for rapid intervention in emergency situations. As-spun multimodal biosensors for the simultaneous and selective detection of body temperature, and pH, $Na^+$, $K^+$, and $Ca^{2+}$ in sweat were fabricated through the use of coaxial wet spinning technology, with silk yarn serving as the core of the biosensors (Supplementary Video 1). These biosensors were then knitted into a multiplexed sensing hairband system in parallel, which possessed a dense bilayer architecture (Supplementary Fig. 2). The interlacing pattern formed by the vertical warp and horizontal weft yarns created multiple interweaving points between the biosensing yarns and the textile substrate, providing both structural reinforcement and positional stability. The adjacent yarns can maintain a certain spacing even under extreme deformation, thus avoiding the short circuit caused by electrode contact (Fig. 1b and c). As-spun biosensors demonstrated rapid water absorption and storage capabilities, attributable to their uniform microporous architecture. The contact angle was measured to be 61°, with complete droplet absorption occurring within 30 seconds, demonstrating superior hydrophilic properties. In contrast, the textile substrate composed of hydrophobic yarns displayed evident non-wetting properties, maintaining incomplete droplet absorption even after 120 seconds (Supplementary Fig. 3 and Supplementary Fig. 4). According to the surface wettability theory, the asymmetric contact angle distribution established a wettability gradient between the biosensing yarns and the textile substrate (Supplementary Fig. 5). The gradient facilitated the directional transport of sweat, thereby enabling precise fluidic capture and regulation of the biosensor array (Fig. 1d). Upon sweat capture, the ionophore-doped biosensors formed a loop system with sweat, converting the activity of ions to potential as output signal. The ionophore demonstrated strong selectivity towards target ions, resisting interference from coexisting ions. It also facilitated charge transfer via electron migration, enabling the charge collection by the carboxylated carbon nanotubes (CNT) within the SF/CNT/PLA (SCP) mixture, which resulted in a measurable shift in the overall interfacial potential (Fig. 1e). Our system collected signals by continuously tracking of both sweat composition and body temperature, thereby enabling real-time health assessment for individuals in emergency conditions.

### Preparation and optimization of as-spun multimodal biosensors

Due to its inherent biological compatibility and biodegradability, silk fibroin (SF) served as the principal substrate component of as-spun multimodal biosensors. However, the SF dispersion exhibited a near-liquid rheological behavior even the solid content was increased to 20%, which was unsuitable for wet spinning due to the lack of viscoelasticity. It is surprising that the dispersion with an unchanged concentration displayed a markedly enhanced viscosity following the mixing of SF, polylactic acid (PLA), and CNT (Fig. 2a and Supplementary Video 2). Rheological measurement confirmed a substantial increase in viscosity from 0.75 to 1277 Pa s at a low shear rate (Fig. 2b). This increase may be attributed to the side groups of amino acids in silk fibroin, which can form van der Waals forces and π-π interactions with carbon nanotubes, thereby enhancing their tight binding[36]. The characteristics absorption peaks of CNT, SF, PLA and SCP mixture are illustrated in Fig. 2c. For CNT, the peaks at 1440 and 1631 $cm^{-1}$ were attributed to O-H bending deformation and -OH stretching vibration, respectively[37,38]. The absorption peak at 1752 $cm^{-1}$ corresponded to the C = O stretching vibration, while the peaks at 1181 $cm^{-1}$ and 1082 $cm^{-1}$ were attributed to the -C = O bending vibration and -C-O- stretching vibration, confirming the presence of ester groups in PLA[39]. Besides, the characteristic absorption peaks for SF were located at 1522, 1630, and 3289 $cm^{-1}$, corresponding to amide II, amide I and -OH stretching vibration peak[40]. The Fourier-transform infrared spectra of the SCP mixture revealed that the characteristic peaks of CNT, PLA and SF do not shift, disappear, or overlap as a result of blending, proving that CNT, PLA and SF are compatible and can be blended effectively.

The ionophores or functional materials were uniformly dispersed within the SCP mixture and then extruded into the coagulation bath through a wet spinning needle. The functional mixture underwent phase separation via the solvent dual-diffusion process, ultimately solidifying into monolithic biosensing yarns with structural uniformity (Fig. 2d and Supplementary Fig. 6). For the SCP-based biosensor fabrication, a single-channel spinning needle was employed to induce direct phase inversion of the functional SCP mixture, resulting in a homogeneous yarn architecture with controlled diameter. In contrast, the core-sheath SCP (CSCP)-based biosensor utilized a coaxial needle with dual isolated microchannels: a central channel for continuous silk yarn feeding and a concentric annular channel for concurrent deposition of functional SCP mixture. Schematic diagrams and cross-sectional SEM images revealed that both SCP and CSCP based biosensors exhibited exceptional structural stability, morphological consistency, and compositional homogeneity. Specifically, the CSCP-based biosensor possessed an optimized core-sheath structure with a central silk yarn, which may provide a certain mechanical support (Supplementary Fig. 7).

To enhance the comprehensive performance, the mechanical, structural, and electrochemical properties of pristine SCP and CSCP yarn were systematically investigated. Stress-strain analysis revealed that CSCP yarns exhibited enhanced mechanical strength compared to SCP counterparts across varying CNT concentrations (Fig. 2e). Statistical quantification demonstrated that CSCP yarns achieved $35.15 \pm 0.7\ MPa$, representing a 17-fold enhancement over SCP yarns ($2.12 \pm 0.5\ MPa$). The considerable enhancement in mechanical strength substantiates the efficacy of the coaxial spinning technology, positioning CSCP yarn as a superior candidate for textile-integrated biosensing applications.

With regard to electrochemical characteristic, CSCP yarns with varying CNT ratios (2-8%) were analyzed in $K_3Fe(CN)_6$/KCl solution to determine the optimal CNT ratio, thus achieving minimized charge transfer resistance and maximized electrochemical activity. Electrochemical impedance spectroscopy demonstrated that the 6% CNT-incorporated CSCP yarn exhibited a reduced semicircle radius, exhibiting minimal charge-transfer resistance among all tested yarns (Supplementary Fig. 8a). Complementary cyclic voltammetry revealed

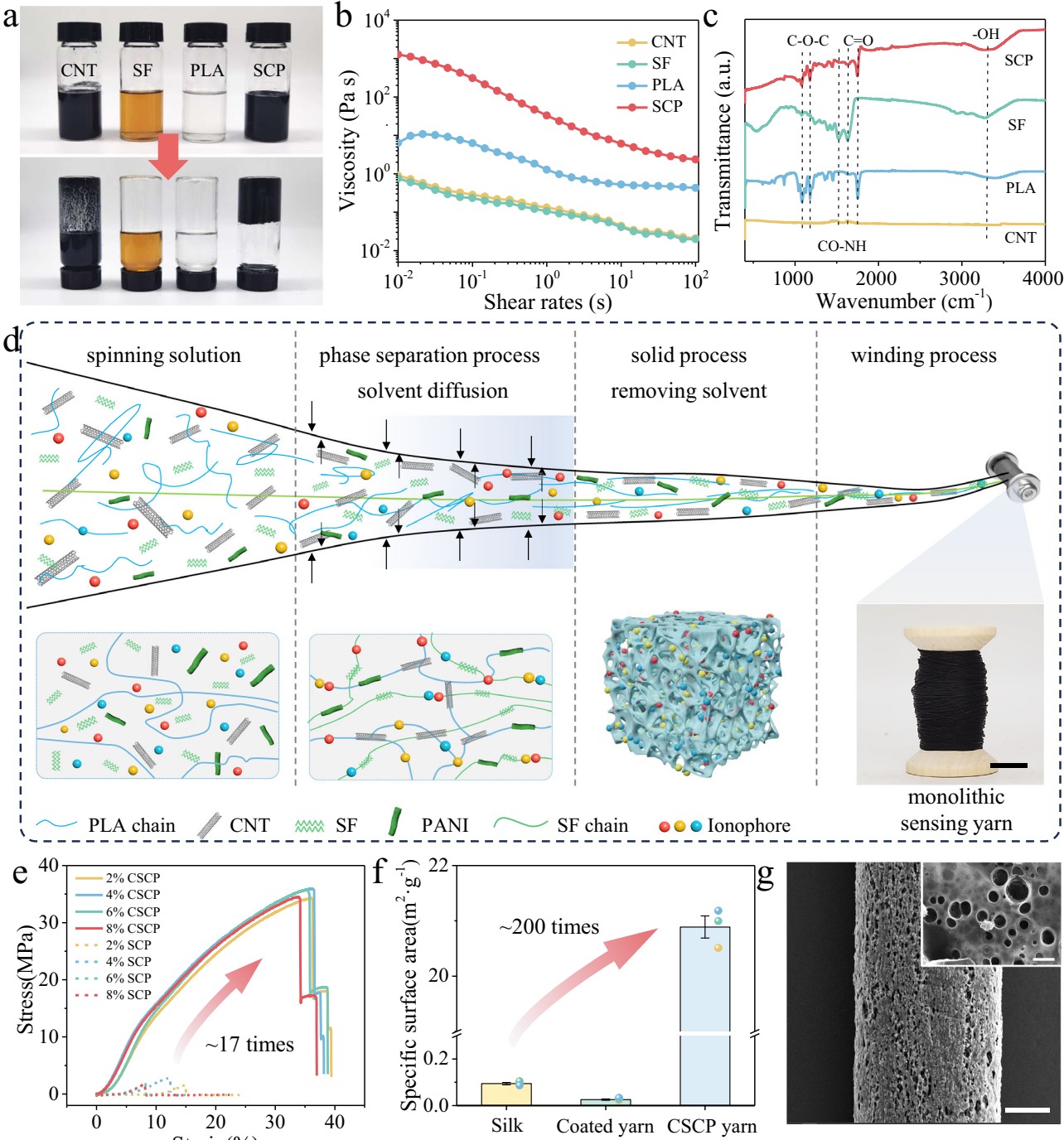

**Fig. 2 | Preparation and characterization of as-spun multimodal biosensors. a–c** Optical images (**a**), viscoelasticity comparison (**b**), and FT-IR spectra (**c**) of CNT, SF, PLA and SCP mixture. SCP, SF/CNT/PLA mixture; CNT, carboxylated carbon nanotubes; PLA, polylactic acid; SF, silk fibroin. **d** Schematic illustration of the preparing monolithic biosensing yarn through wet spinning technique. Inset, a continuously prepared monolithic biosensing yarn. Scale bar, 1 cm. **e** Stress–strain curves of the SCP and CSCP yarns. CSCP, core-sheath SCP. **f** Specific surface area of silk yarn, conventional coated silk yarn, and CSCP yarn (n = 3 independent yarns). Data are presented as mean values + /− SEM. **g** SEM images of CSCP yarn. Scale bars, 200 μm in main image and 1 μm in inset image.

the highest Faraday current[41] and improved capacitive behavior in 6% CNT-incorporated CSCP yarn, suggesting optimized interfacial electron transfer kinetics and surface redox activity. Furthermore, the redox coupling indicated its superior charge storage capacity and electrochemical activity (Supplementary Fig. 8b). These characteristics demonstrated exceptional and stable electrochemical performance of 6% CNT-incorporated CSCP yarn in simulated sweat environment, confirming its suitability for biosensing applications.

Based on this, the reference electrode was fabricated by coating Ag/AgCl conductive slurry onto the 6% CNT-incorporated CSCP yarn and then encapsulating it with the NaCl-saturated PVB membrane. Microstructural analysis confirmed that a continuous Ag/AgCl layer was uniformly adhering to the yarn surface, forming a stable reference electrode interface (Supplementary Fig. 9). The NaCl/PVB membrane was effective in maintaining the reference yarn potential by maintaining Cl⁻ activity and preventing external Cl⁻ interference. The

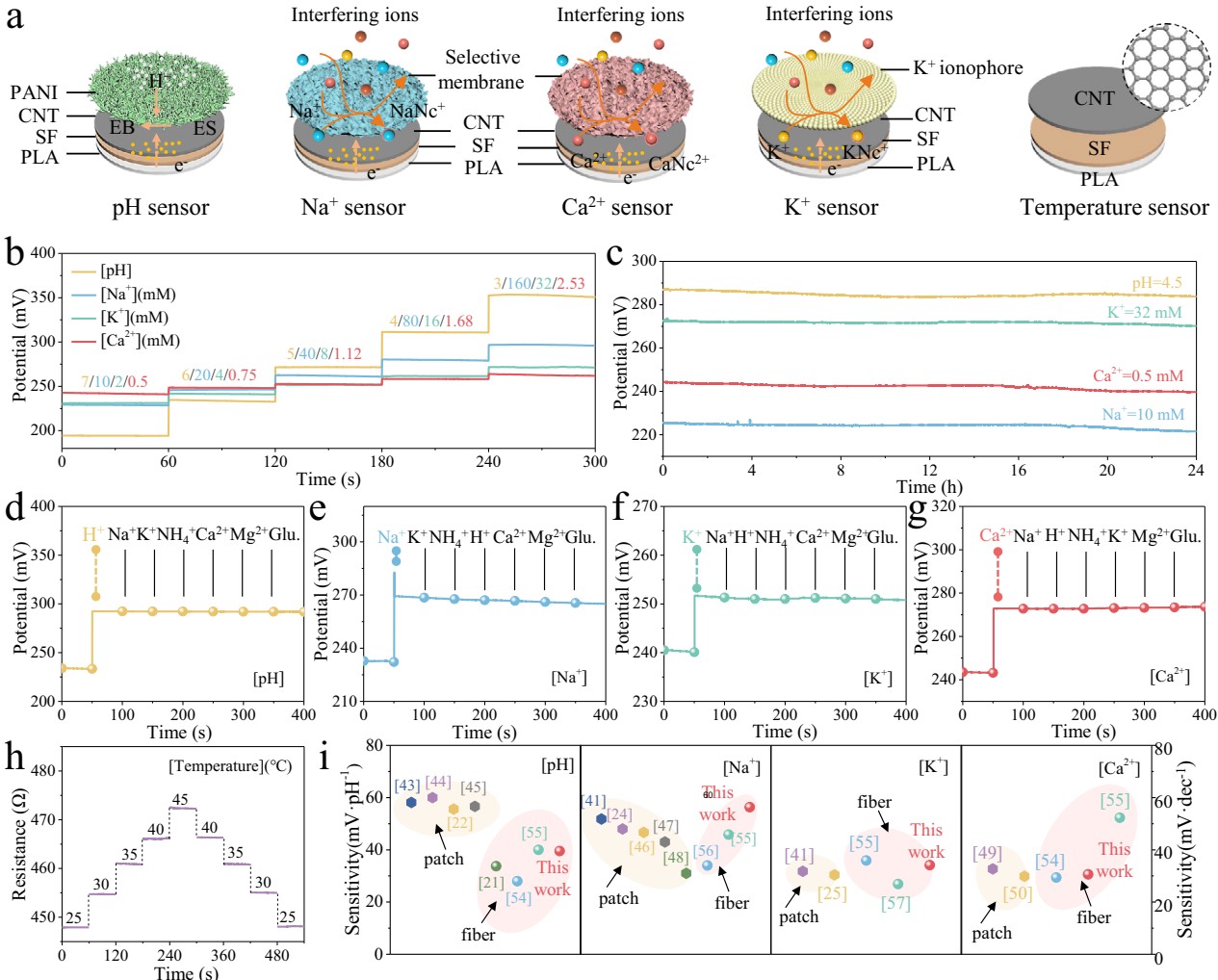

**Fig. 3 | Sensing performance of as-spun multimodal biosensors.**
**a** Configurations and mechanism of pH, Na⁺, K⁺, Ca²⁺ and temperature biosensors.
CNT, carboxylated carbon nanotubes; PLA, polylactic acid; SF, silk fibroin.
**b,c** Potentiometric response (**b**) and long-term stability (**c**) of pH, Na⁺, K⁺ and Ca²⁺ biosensors in PBS solutions. **d–g** Selectivity of pH (**d**), Na⁺ (**e**), K⁺ (**f**), and Ca²⁺ (**g**) biosensors with 1 mM interfering ions. **h** Response of temperature sensor in the physiological temperature range. **i** Sensitivity comparison of as-spun multimodal biosensors with other reported patch or fiber biosensors. Yellow shadings indicate the patch biosensors and pink shadings indicate the fiber biosensors.

potential of the Ag/AgCl yarn was found to be well-maintained under varying salt concentrations (Supplementary Fig. 10).

Furthermore, the structural characteristics of the 6% CNT-incorporated CSCP yarn were investigated using the N₂ adsorption-desorption technique. Compared to the silk core yarn (0.09 m²/g) and the conventional coated silk yarn (0.03 m²/g), the specific surface area of the CSCP yarn was higher at 20.89 m²/g (Fig. 2f). Moreover, the maximum N₂ adsorption capacity of the CSCP yarn was recorded at 98.9 cm³/g, with a corresponding pore volume of 0.16 cm³/g (Supplementary Fig. 11a). Both SEM characterization and pore size distribution analysis revealed that the CSCP yarn featured a porous structure with pore sizes ranging from 200 to 500 nm (Fig. 2g and Supplementary Fig. 11b). This hierarchical porosity not only facilitated efficient sweat absorption but also provided numerous reactive sites for ion and electron transfer.

## Sensing performance of as-spun multimodal biosensors

The configuration and working mechanisms of as-spun multimodal biosensors are schematically depicted in Fig. 3a. The 6% CNT-incorporated SCP mixture is utilized as the current collector for the entirety of the biosensor array, after which different functional materials are selectively incorporated into the matrix to construct pH

biosensors and ion biosensors. Notably, the CSCP yarns are leveraged for temperature sensing due to the exceptional thermal and electrical conductivity endowed by CNT. In the case of pH biosensor, conductive polyaniline (PANI) is employed as the proton sensing membrane. PANI exhibits the unique ability to absorb H⁺ in acidic solutions, which leads to the formation of the conductive emeraldine salt (ES) and causes changes in potential. Conversely, exposure to alkaline solutions results in the release of H⁺ and reversion to the non-conductive emeraldine base (EB)[42]. Ion biosensors utilize the functionalized SCP mixture to selectively bind to target ions, leading to the potential gradient at the interface. The generated electromotive force adheres to the Nernstian equation, allowing for the precise characterization of ion concentration levels in sweat. This design enables the sensitive and selective detection of ions, which is pivotal for accurate sweat analysis and subsequent health monitoring applications.

The monitoring of sweat biomarkers and temperature was highly selective and long-term stable, and a linear correlation was obtained between the potential/resistance responses and the concentrations/states of the targets (Fig. 3b, h). The sensitivities of as-spun pH, Na⁺, K⁺, Ca²⁺, and temperature biosensors were determined through five repeated measurements, yielding values of $39.52 \pm 0.5$ mV pH⁻¹, $56.33 \pm 1$ mV dec⁻¹, $34.13 \pm 0.6$ mV dec⁻¹, $30.61 \pm 0.8$ mV dec⁻¹, and

$1.2 \pm 0.02\ \Omega\ °C^{-1}$, respectively. Calibration curves for all as-spun biosensors were established via linear regression, demonstrating excellent linearity ($R^2 > 0.99$, Supplementary Fig. 12). Furthermore, as-spun biosensors fabricated with the coaxial wet spinning process exhibited high reproducibility with relative standard deviations (RSD) of 1.28% (pH), 2.18% ($Na^+$), 1.78% ($K^+$), 2.67% ($Ca^{2+}$), and 1.87% (temperature), as shown in Supplementary Fig. 13. The durability of as-spun biosensors was also demonstrated by continuous operation in phosphate buffer saline (PBS) solutions containing different analytes for over 24 hours (Fig. 3c and Supplementary Fig. 14). The measured potentials and resistances demonstrated remarkable stability across as-spun biosensors. Specifically, as-spun yarns exhibited a drift of $0.13 \pm 0.01$, $0.17 \pm 0.02$, $0.1 \pm 0.008$, $0.19 \pm 0.01\ mV\ h^{-1}$ for pH, $Na^+$, $K^+$, $Ca^{2+}$ sensing, and $0.05\ \Omega \pm 0.004\ h^{-1}$ for temperature sensing. Therefore, the remarkable long-term stability of as-spun biosensors can meet the requirements for practical applications in the long-term health monitoring environment.

Given the intricate nature of sweat composition, the anti-interference capability of as-spun biosensors was tested by introducing common ions typically present in human sweat into the detection solution. It is noteworthy that as-spun biosensors exhibited a potential fluctuation of less than 1 mV in response to interfering ions of varying concentrations, demonstrating their excellent ability to disregard the impact of common ions (Fig. 3d-g and Supplementary Fig. 15). The long-term stability of as-spun multimodal biosensors was also confirmed through repeated measurements within 60 days, displaying signal degradation rates of 0.81% (pH), 1.43% ($Na^+$), 1.15% ($K^+$), 1.66% ($Ca^{2+}$), and 0.95% (temperature) (Supplementary Fig. 16). It had been observed that ion-selective biosensors exhibited greater response drift, likely induced by a reduction in ionophore activity. A comparison of the sensitivity between as-spun biosensors and previously reported patch[22,24,25,41–53] or textile biosensors[21,54–59] was presented in Fig. 3i and Supplementary Fig. 17. Furthermore, the detection range and long-term stability were discussed in Supplementary Tables 1-5. The comparisons highlighted the high sensitivity and stability of as-spun biosensors within the human physiological range, surpassing that of the majority of existing wearable biosensors in the field.

## Biocompatibility and robustness of as-spun multimodal biosensors

The biocompatibility and robustness of as-spun biosensors have been verified through comprehensive investigation of degradation, cytocompatibility, environmental stability and knittability. As-spun biosensors demonstrated time-dependent surface roughening and localized crack formation, which was attributed to the synergistic effect of enzymatic hydrolysis of silk protein and ester bond cleavage of polylactic acid matrix (Fig. 4a). As illustrated in Fig. 4b, the weight loss rate of as-spun biosensors was measured over the course of the burial degradation process. An escalation in the mass loss percentage of as-spun biosensors was observed in direct proportion to the duration of burial exposure. In particular, after a 30-day interval, as-spun biosensors demonstrated a weight loss rate of around 21%, which escalated to around 68% following a 90-day interval. Collectively, these observations highlighted the significant degradation of as-spun biosensors under burial conditions.

The biocompatibility of wearable biosensor was of paramount importance, particularly in cases where the biosensor was in direct contact with the skin. In vitro safety assessment of as-spun biosensors was conducted utilizing murine fibroblast L929 cells. Following a 72 h co-culture period with as-spun biosensors, fluorescent microscopy images were obtained. Live/dead staining assays revealed minimal cell mortality, indicating that the active materials integrated into the biosensors have no significant impact on cell viability (Fig. 4d). Furthermore, the capacity of L929 cells to proliferate on the as-spun biosensors was evaluated using the CCK-8

experiment. The results demonstrated a linear and exponential growth pattern of cells over 72 hours, thereby confirming the absence of significant cytotoxicity associated with as-spun biosensors (Fig. 4c). The incorporation of PANI and valinomycin resulted in a modest decrease in cell proliferation, but had no substantial impact on cell viability. Univariate analysis of variance showed that the difference of absorbance among different ion groups was statistically significant at 24 h ($P < 0.01$), 48 hours ($P < 0.01$) and 72 h ($P < 0.01$). The standard deviation of the experimental data was less than 0.05, which indicated that the results were repeatable and reliable (Supplementary Fig. 18).

The stability of as-spun biosensors was evaluated under high temperature and humidity conditions using a constant temperature heating table and a custom-built humidity box. During testing, standardized analyze solutions were applied to the biosensors. The multi-biosensor array demonstrated exceptional signal consistency across different environmental conditions, with maximum voltage deviations constrained within 2 mV (Supplementary Fig. 19). Leveraging the intrinsic textile structure and mechanical robustness of as-spun biosensors, the multi-biosensing hairband demonstrated exceptional laundering stability, retaining over 97% initial potential or resistance after 30 laundering cycles (Supplementary Fig. 20). This approach effectively removed sebum secretions, stratum corneum metabolites, and sweat-derived crystalline deposits from as-spun biosensors, thereby maintaining the sensing performance. Furthermore, as-spun biosensors were observed to maintain stability under various deformations, thereby highlighting their superior robustness (Fig. 4e).

Large-scale and continuous production of biosensors is essential for their integration into e-textiles. In the present study, biosensors were processed from the wet spinning technique with a length of hundreds of meters, which can fully meet the needs of the weaving process. As-spun biosensors were then embroidered onto a silk handkerchief, resulting in the formation of a "bear" motif, thereby demonstrating the requisite strength and flexibility for textile manipulation. Furthermore, the incorporation of biosensors did not compromise the lightweight and soft properties of the handkerchief, as evidenced by the waving in the breeze (Fig. 4f). The structure of the biosensing fabric was determined, ensuring that the electrodes were densely packed and reinforced without the risk of electrical short circuits (Fig. 4g). Subsequently, a multimodal biosensing fabric for health monitoring was woven using the SILVER REED SK280 sweater knitting machine (Fig. 4h). The biosensing fabric was then tailored into a multi-biosensing hairband and equipped with a circuit board storage bag to facilitate the on-body analysis (Fig. 4i). The multi-biosensing hairband exhibited optimal air permeability and skin conformability due to its knitted structure, which facilitated effective sweat collection through tight adherence to the skin.

## On-body analysis of the multi-biosensing hairband

To facilitate the practical implementation of the multi-biosensing hairband system, a custom integrated circuit had been engineered, incorporating signal processing units, a Bluetooth module, and a gyroscope sensor, all of which were powered by a battery (Fig. 5a, b and Supplementary Fig. 21). The system architecture involved the acquisition of signals from a biosensor array, which were then input to the circuit board, wirelessly transmitted to a mobile device via the Bluetooth module, and ultimately uploaded to a cloud-based system for data storage and analysis. The integrated gyroscope within the circuit board enabled the monitoring of physical activity, thereby providing a comprehensive overview of the user's health status. A pilot study was conducted with a subject from Qingdao University, who utilized the multi-biosensing hairband system during exercise (Fig. 5c and Supplementary Video 3). The custom-developed user interface allowed for the real-time display of health metrics,

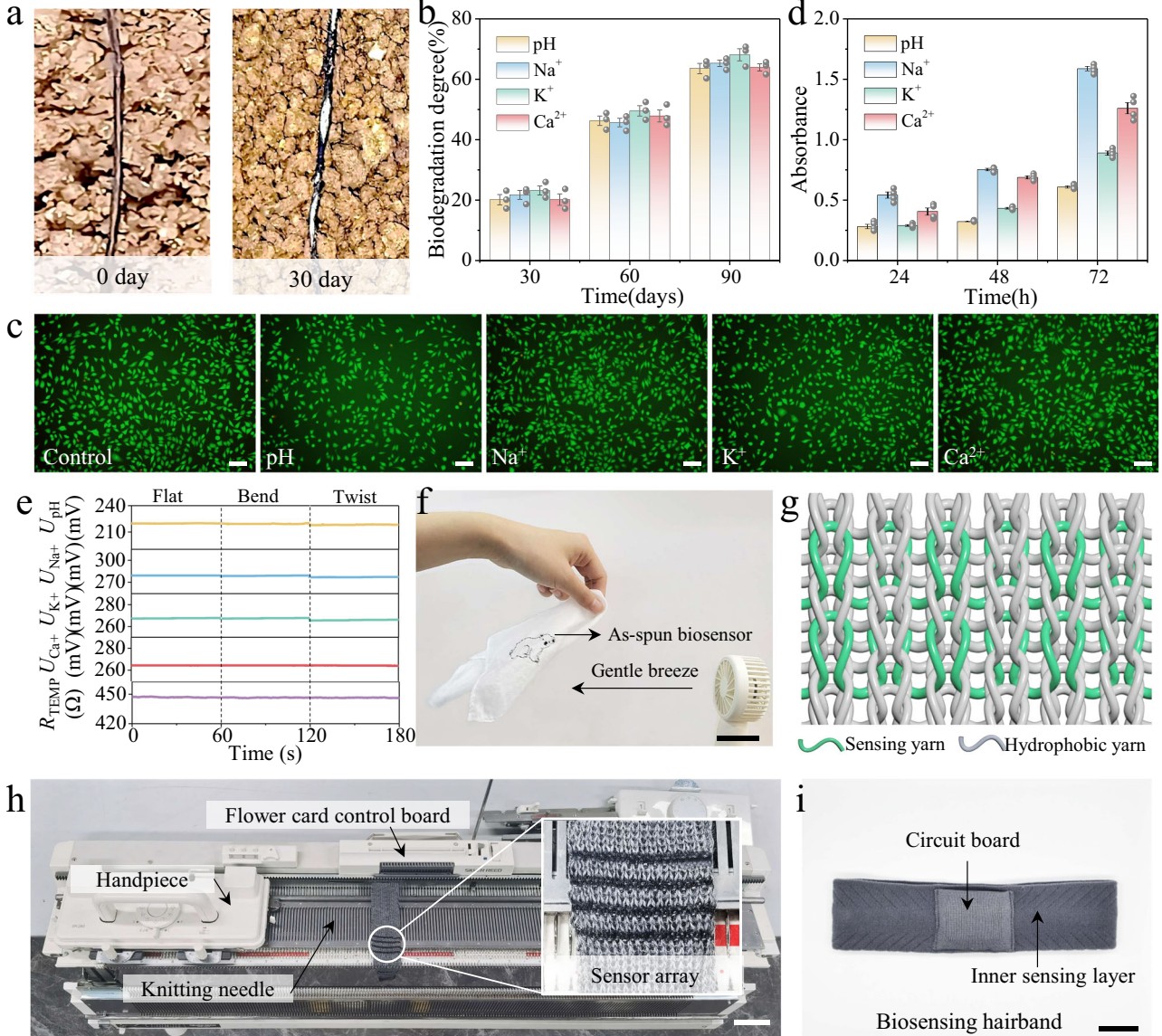

**Fig. 4 | Biocompatibility and robustness of as-spun multimodal biosensors.** **a,b** Optical images (**a**) and weight loss rate (**b**) of as-spun multimodal biosensors during degradation (n = 3 independent biosensors). Data are presented as mean values +/− SEM. **c, d** Fluorescence microscopy images (**c**) and CCK-8 experimental results (**d**) of mouse epithelial cells cultured for 72 h (n = 4 independent experiments). Scale bar, 100 μm. Data are presented as mean values +/− SEM. **e** Stability of as-spun multimodal biosensors under different deformations. **f** Optical image of as-spun biosensor embroidered silk handkerchief under gentle breeze. Scale bar, 5 cm. **g** Schematic of the multi-biosensing fabric structure. **h** Continuous fabrication process of the biosensing fabric. Scale bar, 5 cm. **i** Optical image of the tailored multi-biosensing hairband. Scale bar, 3 cm.

including sweat composition, body temperature, and step count. The practicability and real-time capabilities of the health monitoring system had been verified, demonstrating significant potential for application in emergency health assessments. This study underscored the system's efficacy in capturing critical health data, which can be instrumental in the rapid assessment and management of health emergencies.

As previously reported[60], the distribution density of sweat glands in different parts of the human body was different (Fig. 5d). In order to verify the universality of the findings, the multi-biosensors were affixed to the palm, head and arm to capture physiological signals, with the collected data demonstrating a consistent response across these components of human body (Fig. 5e). Furthermore, five volunteers were also observed wearing the multi-biosensing hairband system during a climbing exercise (Supplementary Table 6). After five minutes of exercise, sufficient sweat was produced on the

multi-biosensors array to obtain a stable output signal. The real-time detection data of six signals related to steps, temperature, pH, Na+, K+ and Ca2+ were shown in Fig. 5f and Supplementary Fig. 22. Throughout continuous exercise, an increase in sweat pH was observed, accompanied by a slight decline in the concentrations of sweat electrolytes.

The analysis of the detected physiological signals was facilitated by the utilization of a box-and-whisker plot, which provided a visual representation of the data distribution across the five volunteers (Supplementary Fig. 23). The measured compositions of sweat fell within the expected normal ranges for healthy individuals, thereby confirming the good health of all five subjects. Following a climbing session lasting 50 minutes, sweat samples were collected to verify the accuracy. The in situ measured concentrations demonstrated a high accuracy rate of over 90% compared to the ex situ measurements, underscoring the reliability and robustness of the

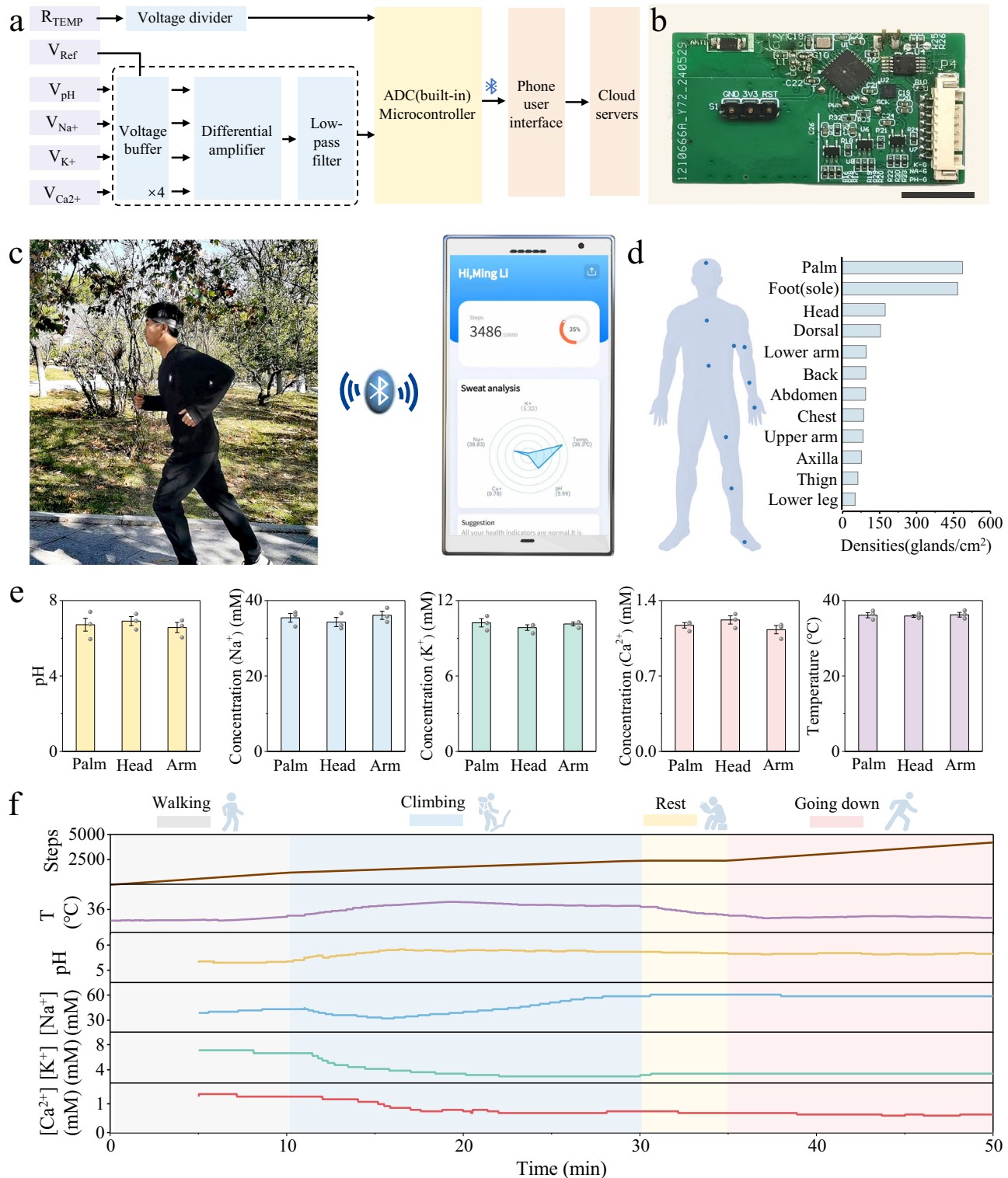

**Fig. 5 | On-body sweat analysis of the multi-biosensing hairband system.**
**a,b** System-level block diagram (**a**) and photograph (**b**) of the circuit patch. Scale bar, 1 cm. **c** A subject wearing the hairband for health assessment. **d** Density of sweat glands in various parts of the human body. **e** Consistency of sweat information from different body locations (n = 3 independent biosensors). Data are presented as mean values ± SEM. **f** Continuous multimodal monitoring of a subject during exercise.

multi-biosensing hairband system for real-time biosensing applications (Supplementary Fig. 24). This technological innovation provided a convenient and robust solution for real-time health assessment, offering significant advantages for the early detection and management of health conditions, leading to more effective interventions in emergency situations.

## Discussion

In summary, a weavable hairband integrated with as-spun multimodal biosensors for real-time physiological signal monitoring in emergency situations is demonstrated. Fabricated through coaxial wet spinning technology, the biosensors feature a seamless, uniform porous structure that facilitates the transport of both sweat and electrons.

Therefore, six signals can be monitored continuously in real time. The sensitivities of pH, $Na^+$, $K^+$, $Ca^{2+}$ and temperature biosensors have been determined to be $39.52 \pm 0.5$ mV $pH^{-1}$, $56.33 \pm 1$ mV $dec^{-1}$, $34.13 \pm 0.6$ mV $dec^{-1}$, $30.61 \pm 0.8$ mV $dec^{-1}$, and $1.2\ \Omega \pm 0.02\ \Omega\ °C^{-1}$, respectively. Furthermore, the minimal signal drift of $0.13 \pm 0.01$, $0.17 \pm 0.02$, $0.1 \pm 0.008$, $0.19 \pm 0.01$ mV $h^{-1}$ for pH, $Na^+$, $K^+$, $Ca^{2+}$ sensing, and $0.05 \pm 0.004\ \Omega\ h^{-1}$ for temperature sensing over 24 hours has been achieved. The utilization of eco-friendly raw materials and the incorporation of designed structures ensure the favorable biodegradability and biocompatibility of the multi-biosensor array, enabling direct prolonged contact with human skin without causing irritation. As a proof-of-concept, the fabricated hairband system incorporates components for signal recording, processing, and wireless transmission, enabling real-time monitoring of human physiological conditions. This innovative design has the potential to be used for telemedicine monitoring and emergency health assessment, and would drive the innovation of the real-time health monitoring technology.

## Methods

### Materials
Silk yarn was provided by Huizhou SiBAINa Textile Co. LTD. Carboxylated carbon nanotube (CNT) and acetic acid were purchased from Tianjin Chemical Reagent Co., Ltd. Silk protein (SF) powder was purchased from HEFEI BOMEI BIOTECHNOLOGY CO.LTD. Glucose anhydrous, polylactic acid (PLA), polyaniline (PANI), Polyvinyl Butyral (PVB), methanol, 1, 4-dioxane, valinomycin, calcium ionophore II and sodium ionophore X were provided by Shanghai Macklin Biochemical Co. Ltd. Phosphate Buffered Saline (PBS, pH 7.2-7.4) was provided by Beijing Solarbio Science& Technology Co., Ltd. Ag/AgCl ink was Shangdong AEX Chemical Technology Co. Limited. Potassium chloride (KCl), sodium chloride (NaCl), calcium chloride ($CaCl_2$), ammonium chloride ($NH_4Cl$) and magnesium chloride (MgCl) were purchased from Sinopharm Chemical Reagent Co., Ltd. L929 cells was purchased from Wuhan Pricella Biotechnology Co., Ltd.

### Preparation of as-spun biosensors
Firstly, 160 mg $ml^{-1}$ CNT and 160 mg $ml^{-1}$ SF were dispersed in 1, 4-dioxane under ultrasonic cell disruptor for 30 min, respectively. PLA was dissolved in 1, 4-dioxane (160 mg $ml^{-1}$) with stirring until complete dissolution was achieved. The two components were then mixed in the required proportion and stirred to obtain a uniform and viscous SCP mixture. Subsequently, the functional SCP mixtures were prepared by mixing SCP mixture with PANI (10 mg $ml^{-1}$), sodium ionophore X (2 mg $ml^{-1}$), valinomycin (3 mg $ml^{-1}$) and calcium ionophore II (2 mg $ml^{-1}$), respectively.

As-spun biosensors were prepared through a self-assembled wet-spinning system, wherein the 16 wt% functional SCP mixture was loaded into a syringe pump and extruded via a needle into a coagulation bath composed of deionized water. Specifically, SCP-based biosensor fabrication utilizes a conventional single-lumen needle (19 G), while CSCP-based biosensor production incorporates a coaxial needle with silk yarn acting as the core (14/19 G). The functional SCP mixture was propelled at a speed of 36 mL/h as the silk yarn was pulled by rollers in the inner channel. The as-prepared temperature, pH, $Na^+$, $K^+$ and $Ca^{2+}$ biosensors were dried at room temperature.

### Preparation of Ag/AgCl yarn
The Ag/AgCl yarn was prepared by coating a layer of Ag/AgCl ink on the surface of CSCP yarn and then drying it at room temperature., Following this, the yarn was encapsulated with a layer of PVB solution containing 79.1 mg PVB and 50 mg NaCl in 1 mL of methanol.

### Degradation experiment
The degradability of as-spun biosensors was determined in a flower bed on the campus of Qingdao University. The ambient temperature of the environment was maintained within the range of 22 °C–28 °C, and the relative humidity was maintained at approximately 85%. Sensing yarns (5 cm in length) were buried at a depth of 5 cm. At predetermined intervals, samples were retrieved for weighing to ascertain the rate of weight loss over time.

### Biocompatibility experiment
As-spun biosensors were sterilized using high temperature and pressure steam (120 °C for 20 min) and added to the medium in 48-well plates. Subsequently, L929 cells were seeded onto the biosensors and cultured in a 5% $CO_2$ incubator at 37 °C for 72 h to allow cell adherence. Following this, the cells were washed once with PBS in order to remove excess serum. For the cytotoxicity experiments, the Live/Dead cell staining solution was added to the plates and left to incubate at room temperature in the dark for 15 min. The staining was terminated by washing three times with PBS, after which the results were observed and photographed at 100X. For the CCK-8 detection assay, the medium was removed after culturing for 24, 48, and 72 h, and each well was washed three times with PBS. 500 µL of CCK-8 solution (10%) was then added to each well and cultured in a 5% $CO_2$ incubator at 37 °C for 2 h. Thereafter, 400 µL of CCK-8 solution was transferred to a fresh 48-well plate, and the absorbance value at 450 nm was measured using a microplate reader. Any mycoplasma contamination was not observed in our cell cultures.

### Characterization
Scanning electron microscope (SEM, ZEISSEVO18) was used to examine the morphologies of biosensing yarns. Optical Contact Angle and 3D Shape Measurement system (DSA100) was utilized to investigate the wettability of the samples. The rheological behavior was characterized by Rotational Rheometer (MCR702). FTIR spectra of the hybrid spinning dope was carried out on Fourier transform infrared spectrometer (Nicolet IS10). The mechanical properties of biosensing yarns were assessed using Universal material testing machine (Instron 5300). The specific surface area was determined by Specific surface area and aperture tester (ASAP-2460) using Brunauer-Emmett-Teller (BET) method (Micromeritics). The sensing performance of temperature biosensor was characterized by a parameter analyzer (Keithley-2450).

### Electrochemical measurements
The electrochemical performance of biosensing yarns was characterized in the two-electrode system by an electrochemical workstation (CHI760e). The CSCP yarn functioned as working electrode, while an Ag/AgCl yarn served as the reference and counter electrodes. 0.01 M PBS with varying concentrations of the analyze was employed as the electrolyte. Electrochemical impedance was tested from 0.1 to $10^5$ Hz by A.C. impedance program. Cyclic voltammetry curves were performed in a 5.0 mM $K_3Fe(CN)_6$/0.1 M KCl solution, with a scan range from −0.1 to 0.6 V at a scan rate of 5 mV/s. The pH, $Na^+$, $K^+$ and $Ca^{2+}$ sensing yarns were recorded by open-circuit potential measurement in the corresponding analyte solutions. All the electrochemical performances were repeated at least five times. All the data were analyzed through Origin 2015.

### Ethics
This study was approved by the Ethical Committee of the Affiliated Hospital of Qingdao University (QYFYWZLL30286). Five healthy volunteers (two females and three males, Supplementary Table 6) aged from 23 to 30 years were recruited from Qingdao University and each

participant gave informed written consent. Sex and gender information was only collected and shown as overall numbers for statistical information, which is irrelevant to the research findings. The sex and self-identified gender showed no discrepancies among the participants involved in this study. The study was fully voluntary, and no compensation was given. The human research participants approved for the publication of the collected images in Figs. 1b, 5c, as well as the movies in Supplementary Videos 3.

## On-body sweat analysis

The validation and evaluation of the multi-biosensing hairband were performed on recruited subjects during a constant climbing exercise. All subjects fasted overnight before participating in the on-body sweat analysis. The multi-biosensing hairband was worn on the subjects' heads after their skin was cleaned with alcohol wipes. During climbing, the real-time monitoring data was transmitted wirelessly to a mobile phone via Bluetooth.

## Off-body sweat analysis

Post-exercise sweat samples were collected from the same skin area over which the hairband had been positioned for off-body data validation. For ion-sensing results validation, 2 µL sweat was applied to another biosensor array, and the subsequent potential responses were monitored using an electrochemical workstation (CHI760e). Ionic concentrations were derived through linear regression analysis ($y = kx + b$) utilizing standard calibration solutions, where y represents the measured potential and x denotes the logarithmic ion concentration. The pH values were verified using a pH meter (LICHEN PH-10), while temperature measurements were acquired using a thermometer (WDKL-EWQ-004).

## Reporting summary

Further information on research design is available in the Nature Portfolio Reporting Summary linked to this article.

## Data availability

All data supporting the findings of this study are available within the article and its supplementary files at https://doi.org/10.6084/m9.figshare.29465285. Any additional requests for information can be directed to, and will be fulfilled by, the corresponding authors. Source data are provided with this paper.

## Code availability

Custom code is available at https://github.com/Liming-dot/Multi-biosensing-hairband.

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

## Acknowledgements

This work was supported by National Natural Science Foundation of China (52473307, M. T.), Taishan Scholar Program of Shandong Province in China (tsqn202211116, M. T.), Natural Science Foundation of Qingdao (23-2-1-249-zyyd-jch, M. T.), Qingdao Key Technology Research and Industrialization Demonstration Projects (23-1-7-zdfn-2-hz, M. T.), Qingdao Shinan District Science and Technology Plan Project (2022-3-005-DZ, M. T.).

## Author contributions

M.L., N.Y., and M.T. conceived and planned the experiments. M.L., G.L., T.X., R.X., and X.Z. contributed to sample preparation, measurements, and formal analysis. M.L., T.X., and F.C. designed and performed on-body sweat analysis. M.L. and Y.W. writing original draft. M.L., G.L., N.Y., and M.T. reviewed and edited the manuscript. M.T. supervised the project. All authors helped shape the research and revise the manuscript.

## Competing interests
The authors declare no competing interests.
