## [Transparent Peer Review file · Nature Communications]

Multi-Biosensing Hairband for Emergency Health Assessment

Corresponding Author: Professor Mingwei Tian

Version 1:

Reviewer comments:

Reviewer #1

(Remarks to the Author)

The authors demonstrate an ultrasensitive and stable multi-biosensing hairband system based on coaxial wet spinning technology with the creation of seamless and uniform interpenetrating interfaces within the sensors. The resulting biosensing yarns presenting an approximately 200 times increase in specific surface area, significantly improving sensor performance. This work exhibits high innovation and application potential, and could have broad implications for biosensors for real-time physiological signals detection. It is interesting, and could be considered for acceptance after a minor revision. A few suggestions were provided as below.

1. The paper emphasizes that coaxial wet spinning technology improves the sensitivity and stability of the sensor, but does not compare the specific performance differences (such as detection range, long-term stability, etc.) of existing sweat sensors in depth. It is recommended to supplement the table or data comparison to clarify the technical advantages.
2. The distribution statistics of the spun yarn sensor pore structure (200-500 nm) are missing. It is suggested to supplement the aperture distribution histogram.
3. The article mentions that "interfering ions cause potential fluctuations of less than 1 mV", but does not specify the specific concentration range of interfering matter. The concentration gradient data of the interference experiment should be supplemented to verify the selectivity.
4. Although the article mentions that the sensor remains stable within 5000 seconds, practical applications require consideration of continuous use performance over longer periods. It is recommended to supplement long-term stability test data (e.g. 24 hours of continuous monitoring results).
5. In the biocompatibility experiment (Fig. 4c-d), only "n=4 independent experiments" is mentioned, but statistical methods of data processing are not explained. It is suggested to supplement statistical analysis to ensure the reliability of experimental results.
6. The performance of biosensors also needs to be tested in extreme environments such as high temperature and humidity.
7. Some terms are used inconsistently (e.g., "sweat acquiring" vs. "sweat capture"), and there are a few syntax errors. It is suggested to unify the terminology and polish the language throughout the article.

Reviewer #2

(Remarks to the Author)

The article by Ming Li et al. discusses the development of a smart hairband designed for monitoring various biochemical and biophysical parameters. The research topic is highly relevant, and the results presented in the article are promising, although some aspects require clarification. However, the presentation of the results needs improvement to meet the standards necessary for publication in Nature Communications. In particular, the authors should emphasize the novelty of their work while maintaining a high scientific rigor throughout the manuscript. For this reason, the article is not publishable in its proposed format; it could only achieve the required level for publication after significant revisions.

Main comments:

- 1) The introduction should provide background on the recent literature regarding wearable sensors, with a particular focus on textile sensors. While the authors effectively explain the general motivations behind the development of these sensors, they do not discuss the technical advancements necessary to achieve the Technology Readiness Level required for commercialization. What type of transduction is primarily used in the design of wearable sensors, and why? What materials are predominantly utilized in the fabrication of wearable sensors? What configurations are employed in the creation of

wearable and textile-based electrochemical sensors? It is essential to mention amperometric, potentiometric, and transistor-based designs. A comparison of the results with recent literature would help better contextualize the device's performance. What innovative aspects does this research present compared to the current state of the art? All these points should be thoroughly discussed. Additionally, it would be beneficial to cite the following works: 10.1038/nature16521 and <https://doi.org/10.1038/s41598-020-74337-w>. All this information must be integrated into the text.

2) The structure of the wearable sensor needs to be described and discussed in greater detail, particularly regarding the reference electrode. In the wearable sensor, the reference electrode is a strip of Ag/AgCl, whereas in sensor characterization, a traditional electrode is used. The potential of Ag/AgCl depends on the Cl⁻ concentration of the solution in which it is immersed, making it crucial to study potential interferences with the final device. The authors should provide a more comprehensive description of the device's structure while identifying how the presence of Cl⁻ interferes with measurements. All interference tests must be conducted using the final device, and the interference from chlorides should be discussed in detail.

3) Fiber characterization and fabrication. Why do the authors state that the fiber are core-sheath structured? The authors should provide experimental evidence of it. The text is not clear about the difference between CSCP and SCP. This difference should be discussed describing the fabrication procedures and discussing the experimental results. Finally the electrochemical characterization should be enlarged in the main text.

4) Analytical concerns of proposed sensors. The directional transport of sweat is a key point of the proposed device is a key point of proposed device because it is the sampling system that guarantee the continuous monitoring of the human perspiration. The authors discuss this point at lines 102-109. A more detailed discussion (and more characterizations) must be added in the text in order to provide the reader the information about this important point of the proposed device. The compounds (also the analytes) that are present on the skin wearer can interfere with the measurement. How does the authors avoid this problem? The authors describe the operating principles of the sensors in lines 194 – 197. This discussion should be followed the literature concerning potentiometric sensors. The sensors does not used ion channel (no membrane protein has been added to the sensors). The potential change is proportional to the concentration (activity) logarithm and not to the binding strength, as demonstrated by the calibration plots.

5) The title should be changed. Carrying out the experiments in an ambulance setting does not provide valuable evidence considering emergency use. Moreover, the sensors responses are close to the response of common potentiometric sensors. Therefore, the sensors are not ultrasensitive.

Other comments that requires a text modification:

Abstract: abstract should me more quantitative and informative

Introduction. Sweat analysis. the weak point of sweat analysis should be discussed in the text

Introduction and/or results: Spinning technology. Spinning technologies should be explained in order to provide the reader useful information about the fabrication technology.

Lines 320-322: authors should provide more experimental detail concerning the validation of proposed sensors. The instrumentation and the analytical procedure should be added in the experimental section.

Minor comments:

Line 14: which are the biomarkers?

Line 18: "electron transmission" is misleading in the field of electrochemical sensor. It is related to TEM

Introduction:

Line 35: blood gas analyzer and biochemical analyzer should be replaced with a little discussion concerning the gold standard techniques of blood analyzes

Lines 49-51: references should be added

Lines 51-53: references should be added

Lines 57-59: The technology transfer from PET and PDSM to textile fibers needs to be discussed.

Line 62: references should be added

Lines 63-64: references should be added

Lines 79-81: these data are not appropriate for introduction section

Line 185: how can a scp solution be used as current collector?

Line 204: the Ca⁺⁺ slope is 73.64. this value is higher than the Nernstian slope of 29 mV decade⁻¹. The author should provide a detailed explanation of it.

Line 297 figure 5: ambulance setting should be removed.

Reviewer #3

(Remarks to the Author)

The paper "Ultrasensitive and Stable Multi-Biosensing Hairband for Emergency Health Assessment" from M. Li et al reports a multi-analytes sensors realized using coaxial wet spinning technology, having silk yarn as the core. The as-spun sensors proves to have the target sensitivity range, flexibility, stability and capability of being integrated into a wearable hairband. The final configuration is also tested during exercise proving the achievement of real-time physiological analysis.

Despite the work is interesting for the Nature Communication readers community and reaches a high technological level, the article still needs some investigations and several refinement before publication. The main suggestions and comments are reported here below:

1) Page 1, Abstract: The authors should specificity the biomarkers and the range they monitor in the abstract.

2) Page 4, "The drifts of the as-spun multimodal sensors are 1.87, 1.36, 2.3, 0.14 mV h⁻¹ for pH, Na⁺, K⁺, Ca²⁺ sensing, and 0.43 Ω h⁻¹ for temperature sensing,". Instead of focusing on the drift of the sensors, I would rather insert their sensitivity (eventually with th comparison of this drift of the baseline if needed).

- 3) Page 5, "Subsequently, as-spun multimodal sensor is knitted into a multiplexed sensing hairband system in parallel, thus effectively avoiding the short circuit caused by electrode contact": What do the authors mean by knitting in parallel to avoid short circuit? This description and fabrication process is not so clearly reported.
- 4) Page 6, "the interfering ions, leading to the potential change at the total interface": The authors should better explain this mechanism. What is the potential change? What are the interfaces that are involved in this change and how it occurs?
- 5) Page 8, "SF/CNT/PLA (CSCP)": The authors should add a cross-section figure to better explain the structure of the final core-sheath structured yarn.
- 6) Page 8, "stress-strain curves": It looks like there are no differences among the yarns having different percentage of CNT. Is it the case? Can the authors comment on that and explain why the stretchability/flexibility does not depend on CNT concentrations?
- 7) Page 8, "breaking strenght of 35.95 MPa": It is not clear which is the curve the authors refer into the graph, since there are different curves having diverse percentages. What are those numnbers referred to? If multiple tests have been done and some statistical analysiss has been carried out for these breaking strenght tests, the authors should provide the average values and the absolute error associated.
- 8) Page 12, "The sensitivities of the pH, ... 1.21 Ohm C-1": The authors should add the standard deviation of the sensitivities here reported. the Supporting Info should also report the complete fitting equation, with the intercept as well.
- 9) Page 12, "corresponding solution": What is the corresponding solution? Is it sweat, artificial sweat or simply a buffer solution?
- 10) Page 12, "reproducibility": The authors should calculate the reproducibility percentage or the reproducibility range, highlighting the deviation from the "standard"(average) behaviour that could be found in the sensors. Similarly, the authors should calculated the long-term stability for consecutive calibrations, better describing the normalized intensity that is reported in Supp Fig 9. What is this parameter and what is the amount of the decreased response?
- 11) Page 12, "previously reported patch or fiber": Why there is no comparison for the temperature sensor?
- 12) Page 13-15, Figure 4a: In Figure 4a the authors said the degradation analysis was done onto sensors: what is the sensor tested reported in the image? Are there differences among sensors into their degradation? Have the authors tested also their performances in terms of sensitivity and limit of detection? What are the white circles in the images? Is there a better image or higher resolution the authors can use? This looks very blurred.
- 13) Page 14, "Following a 24-hour co-culture period": Why did the authors take the fluourescent images after 24h and not after 72h in order to have the parallel evaluation as the CCK-8 experiment? The authors should add these timings in the Figure caption.
- 14) Page 14, Fig.4d: The authors should name the figure in the order they are cited, so this would be Fig. 4c.
- 15) Page 18, "Following a 50-minute climbing session, sweat samples were collected to verify the accuracy": It is not clear if the comparison (for accuracy) have been done using the same as-spun sensors ex situ or a different as spun sensor or an analytical technique as validation. If no validation has been carried out, tha authors should think of validating the sensors using a gold standard technique to assess these parameters.
- 16) Page 20, Conclusion: The authors should report some of the quantitative findings of the paper in the Conclusion for readers that do not read the whole manuscript but still need some detailed information, such for example the sensitivities of the sensors.

Version 2:

Reviewer comments:

Reviewer #1

(Remarks to the Author)

All my previous concerns were well addressed in a point-to-point way. The presentation of the results and the novelty is improved comparing with the origin version. Thus, I would like to recommend it for acceptance.

Reviewer #2

(Remarks to the Author)

The manuscript can be published in the current form

Reviewer #3

(Remarks to the Author)

The authors strongly improved the initial version of the manuscript and answered to all the Reviewers' comments. However, there are still some minor revision I would suggest before publishing their results.

In particular, the minor concerns and adjustments are the following:

- 1) Abstract & Conclusion: The authors should report the sensitivities with the associated error; the same issue occurs at the end of the Introduction section. The same holds for the drift values that are reported without the errors.
- 2) Errors should be reported using a single digit: this problem occurs throughout the whole manuscript.
- 3) "Fig. R2 Sensing mechanism of potentiometric biosensors in sweat" Are the authors sure that the electrons derive into the silk fiber? My guess is that electrons are collected by the conductive carboxylated carbon nanotubes present in the SCP mixture, thus changing the electrode/material potential. This needs to be clearly reported in the main text as well.
- 4) "Standard solutions": the authors should state and report what they mean for standard solution. Is it Phosphate Buffer

Saline?

5)“Electron transmission”: The authors should use “electron transport” instead, since it is misleading as electron transmission may refer to imaging techniques

1. Figure 3 is not referenced in the text. Fig. 3 contains important info about the calibration and without a proper description, I can hardly understand it. Captions and legends do not help understanding.

Response: Thank you for your helpful comment. We extend our sincere apologies for the error in the image number. Figure 3 has been accurately referenced and expressed in the correct location, and the corresponding explanatory text has been further improved in the latest manuscript.

2. Figures have many mistakes or elements poorly described in the legends and in the captions, both in the SI and in the Main.

Response: Thank you for your helpful comment. We extend our sincere apologies for the error in the legends and captions of Figures. All the legends and instructions in SI and Main have been further revised to make the article more rigorous and formal.

3. you mention experiments on 5 subjects but report only one. Please report your results on humans in full with appropriate statistical analysis.

Response: Thank you for your helpful comment. Five healthy volunteers, aged between 23 and 30 years, were recruited from Qingdao University to participate in the on-body analysis using multi-biosensing hairband. Detailed demographic information for each participant was provided in Table R1.

Table R1. Personal information of five volunteers participating in the on-body sweat analysis using a multi-biosensing hairband.

Subject	Age	Sex	Height(cm)	Weight(kg)
#1	23	Male	178	75

#2	27	Female	158	50
#3	25	Male	176	67
#4	28	Female	165	62
#5	30	Male	176	78

During the exercise, data collected from the subjects were continuously recorded and presented in Figure R1. Increased blood circulation and elevated energy expenditure resulted in variable increases in body temperature during the climbing phase. As the as-spun sensors absorbed sufficient sweat, the pH value and concentrations of Na^+ , K^+ , and Ca^{2+} were consistently transmitted through the integrated monitoring system in real-time. Throughout the continuous exercise, sweat pH increased, while the concentrations of sweat electrolytes exhibited a slight decline. These results demonstrated the potential of the multi-biosensing hairband for emergency health assessment.

Figure R1. On-body multimodal monitoring for health monitoring during exercise. a-e Multimodal sensor responses in four healthy subjects using the multi-biosensing hairband during exercise.

Throughout the real-time monitoring period, the multi-biosensing hairband successfully recorded parameters including pH, Na⁺, K⁺, Ca²⁺, and temperature. To analyze the detected physiological signals, we employed a box-and-whisker plot to illustrate the data distribution across the five volunteers (Figure R2). Based on previous studies (1-3), the normal concentration ranges for these five biomarkers in sweat were indicated by the dashed lines. Notably, the body temperatures of female participants were slightly higher than those of male participants. The measured sweat compositions fell within the expected normal ranges for healthy individuals, thereby confirming the good health of all five subjects.

Figure R2. Distribution of physiological signals collected from five subjects during exercise. a-c Physiological signals of pH (a), Na⁺ (b), K⁺ (c), Ca²⁺ (d), and temperature (e) collected using the multi-biosensing hairband. The dashed lines indicate the upper and lower limits of the normal range for the biomarkers. The box ends represent the 25th and 75th percentiles. The horizontal line in each box represents the median. The

upper and lower whiskers represent the maxima and minima, respectively, which refer to the range of non-outlier data values.

1. Baker, L. B. & Wolfe, A. S. Physiological mechanisms determining eccrine sweat composition. *Eur. J. Appl. Physiol.* **120**, 719-752 (2020).
2. Patterson, M. J., Galloway, S. D. R. & Nimmo, M. A. Variations in Regional Sweat Composition in Normal Human Males. *Exp. Physiol.* **85**, 869-875 (2000).
3. Choudhury, S., Deepak, D., Bhattacharya, G., McLaughlign, J. & Roy, S. S. MoS₂-Polyaniline Based Flexible Electrochemical Biosensor: Toward pH Monitoring in Human Sweat. *Macromol. Mater. Eng.* **308**, 2300007 (2023).

4. Where do you report the ex situ testing of pH?

Response: Thank you for your helpful comment. We have supplemented the ex situ measurements of pH, Na⁺, K⁺, Ca²⁺, and temperature in supplementary information, and the results are described in detail as follows:

Following a 50-minute climbing session, sweat samples were collected from the same skin area where the sensors were worn and then subjected to ion-sensing testing using an electrochemical workstation. The pH values were verified using a pH meter, while temperature measurements were cross-checked with a thermometer. As shown in Figure R3, the in situ measured concentrations demonstrated a high accuracy rate of over 90% compared to the ex situ measurements. This result underscored the reliability and robustness of the multi-biosensing hairband system for real-time biosensing applications.

Figure R3. Ex situ measurements of physiological signals collected from five subjects during exercise. a-c Comparison of (a) pH, (b) Na⁺, (c) K⁺, (d) Ca²⁺ concentrations, and (e) temperature detected by the multi-biosensing hairband and standardized measurement.

5. How do you change the temperature during calibration. Changes in Resistance are unusually fast.

Response: The temperature sensitivity of the as-spun yarn was evaluated in a constant temperature oven (LICHEN YLE-2000) whose operating principle was based on heat conduction, convection, and radiation, thereby ensuring efficient and uniform heating of the samples. Specifically, the oven temperature was initially set to 25 °C, and the sensing yarn was placed into the oven once the temperature had stabilized. Subsequently, a parameter analyzer (Keithley-2450) was connected to the yarn for resistance testing. After the initial test was completed, the oven temperature was adjusted to the next level. These steps were then repeated to complete the sensing

performance test of the temperature sensor over the physiological temperature range of 25 to 45 °C.

Figure R4. Schematic diagram of the temperature sensor test system containing temperature controller, parameter analyzer, and data acquisition module. Scale bar, 1 cm.

It is hypothesized that the misinterpretation may arise from the lack of rigor in the graphical representation. Therefore, the solid lines used for connections in Figure 3h have been replaced with dashed lines. This modification aims to provide a more precise and standardized visual presentation of the data, thereby facilitating a clearer understanding of the results.

Figure R5. Response of the temperature sensor in the physiological temperature range.

6. I suggest you improve the overall quality of the main text, develop figures, captions

and legends, and provide a video of a user putting on the band and recording with the app.

Response: Thank you for your helpful comment. We have supplemented a video of a user putting on the band and recording with the app as a separate file.

Response to Reviewers for NCOMMS-25-01635

Dear Reviewers,

Thank you for your consideration of our manuscript, “**Multi-Biosensing Hairband for Emergency Health Assessment**” for publication in *Nature Communications*. We also extend our gratitude to the Reviewers for their insightful and helpful comments. Below, we have provided a detailed response to each item mentioned by the Reviewers and indicated how we have incorporated their suggestions into the revision.

We would like to summarize the responses to the Reviewers, and the updates made to the manuscript as follows:

1) **Additional Details:** Per the Reviewer’s constructive suggestions, we have added relevant descriptions for detailed demonstrations, including the comprehensive sensing performance comparison (Supplementary Tables 1-5), directional sweat transportation (Supplementary Fig. 4 and Fig. 5), the fabrication and structure characterization (Supplementary Fig. 6 and Fig. 7), and the validation of proposed biosensors. We have also modified the Introduction, Abstract, and Conclusion to be more technical, quantitative, and informative. Finally, we have corrected some errors, such as blurred images, ambiguous expressions, and so on. We have highlighted all the revisions in blue so that the Reviewers can comment on them after our revision.

2) **Extended System Performance Validation:** Per the Reviewer’s valuable suggestions, we have conducted additional experiments to verify the sensing performance in terms of selectivity (Supplementary Fig. 15), long-term stability (Fig. 3c), and robustness (Supplementary Fig. 19 and Fig. 20). We have also supplemented the fluorescent images of mouse epithelial cells by co-culturing murine fibroblast L929 cells with the as-spun biosensors for 72 hours (Fig. 4c). We have highlighted all the corresponding discussion in blue so that the Reviewers and editorial group can comment on them after our revision.

Thank you again for your further consideration of the manuscript for publication in *Nature Communications*.

Best Regards,

Mingwei Tian, Professor

Email: mwtian@qdu.edu.cn

List of detail changes:

1. Page1: Abstract has been modified to be more quantitative and informative.
2. Page 2-5: Introduction has been reconstructed according to the reviewer's suggestion.
3. Page 6: A discussion on the knitted structure to avoid short circuit has been added.
4. Page 6-7: An explanation for the directional transport of sweat has been added.
5. Page 7: The sensing mechanism has been explained in detail.
6. Page 9: The description for the fabrication and structure characterization of as-spun biosensors has been added.
7. Page 10: The electrochemical characterization has been expanded.
8. Page 11: An explanation for the anti-interferes of Cl^- has been added.
9. Page 13: The ion-sensing mechanism has been provided.
10. Page 13-14: The sensing performance of biosensors has been analyzed in depth.
11. Fig. 3c has been updated by measuring stability over 24 hours.
12. Page 15: A comprehensive comparison between as-spun biosensors and previously reported biosensors has been added in Supplementary Tables 1-5.
13. Fig. 4a has been replaced with the high-quality images.
14. The figure numbers annotated for Fig. 4c and Fig. 4d have been corrected.
15. Fig. 4c has been updated by co-culturing murine fibroblast L929 cells with the as-spun biosensors for 72 hours.
16. Page 17: A discussion on the statistical analysis of biocompatibility experiment has been added.
17. Page 17-18: The robustness of as-spun biosensors has been discussed according to the reviewer's suggestion.
18. Fig. 5c and its caption has been changed according to the reviewer's suggestion.
19. Page 23: Conclusion has been modified to be more quantitative and informative.
20. Page 24-25: The preparation of as-spun biosensors has been explained in detail.
21. Page 25: The preparation of Ag/AgCl yarn has been updated.
22. Page 27: The validation of proposed biosensors has been explained in detail.
23. According to the revisions, the following figures has been updated or added:
Supplementary Figs.1, 4-7, 9-11, and 14-20.
24. The list of references has been updated.

Reviewer #1

Comments: The authors demonstrate an ultrasensitive and stable multi-biosensing hairband system based on coaxial wet spinning technology with the creation of seamless and uniform interpenetrating interfaces within the sensors. The resulting biosensing yarns presenting an approximately 200 times increase in specific surface area, significantly improving sensor performance. This work exhibits high innovation and application potential, and could have broad implications for biosensors for real-time physiological signals detection. It is interesting, and could be considered for acceptance after a minor revision. A few suggestions were provided as below.

Response: We appreciate the positive evaluation and affirmation of our research by the reviewer, and we thank you for the recommendation to publish in *Nature Communications*. Below, we respond to each point raised individually by the Reviewer and discuss how we have incorporated the reviewer's suggestions into the revision.

1. The paper emphasizes that coaxial wet spinning technology improves the sensitivity and stability of the sensor, but does not compare the specific performance differences (such as detection range, long-term stability, etc.) of existing sweat sensors in depth. It is recommended to supplement the table or data comparison to clarify the technical advantages.

Response: Thank you for the constructive remark. We have conducted a comprehensive comparative analysis of the as-spun multimodal biosensors (pH, Na⁺, K⁺, Ca²⁺ and temperature) with respect to their substrate, sensitivity, detection range, and long-term stability. The results demonstrate that the as-spun multimodal biosensors exhibit superior performance in both sensitivity and long-term stability within the human physiological range.

In response to reviewer's comment, we have added new Supplementary Tables 1-5 in the revised supplementary information, along with a related description in the revised manuscript (highlighted in blue).

Supplementary Table 1. Comparison of the pH sensing performance

Substrate	Detection range	Sensitivity (mV pH ⁻¹)	long-term stability (hours)	Ref.
PVDF nanofiber	3.7-8.5	30.1	-	1
PVDF&PAN	3-8	29.3	-	2
CNT fiber	4-7	28	10	3

PDMS fiber	4-8	33.75	-	4
CNT fiber	4-7	38.3	1.39	5
Blend cloth	4-10	40±4	1	6
Silk/PLA/CNT yarn	3-7	39.52	24	This work

Supplementary Table 2. Comparison of the Na⁺ sensing performance

Substrate	Detection range (mM)	Sensitivity (mV dec ⁻¹)	long-term stability (hours)	Ref.
CNT fiber	10-160	45.8	1.39	5
PVDF	10-160	31.3	-	7
PET	10-160	43.1	2	8
PDMS	5-80	46.66	0.56	9
PET	5-100	51.8	-	10
PI	10-160	48	-	11
Silk/PLA/CNT yarn	10-160	56.33	24	This work

Supplementary Table 3. Comparison of the K⁺ sensing performance

Substrate	Detection range (mM)	Sensitivity (mV dec ⁻¹)	long-term stability (hours)	Ref.
CNT fiber	2-32	39	10	3
CNT fiber	2-32	35.9	1.39	5
PET	1.25-40	31.8	-	10
PI	0.625-40	30.42	3	12
TANi/GO fiber	2-32	26.9	-	13
PAN/PVP	0-32	34.7	1.67	14
Silk/PLA/CNT yarn	2-32	34.13	24	This work

Supplementary Table 4. Comparison of the Ca²⁺ sensing performance

Substrate	Detection range (mM)	Sensitivity (mV dec ⁻¹)	long-term stability (hours)	Ref.
CNT fiber	4-8	29.4	10	3
CNT fiber	0.5-2.53	52.3	1.39	5
PET	0.25-2	32.7	4	15
Circuit boards	10 ⁻⁴ -100	29.91	24	16

PVC	0.1-100	30	-	17
LIG	0.015-10	27.7	-	18
Silk/PLA/CNT yarn	0.5-2.53	30.61	24	This work

Supplementary Table 5. Comparison of the temperature sensing performance

Substrate	Detection range (°C)	Sensitivity (% °C ⁻¹)	long-term stability (hours)	Ref.
Taffeta fabric	25-50	-1.04		19
PI	28-90	0.113	-	20
Kapton	20-60	0.204	-	21
Taffeta fabric	25-50	0.31	-	22
PI	30-100	0.05		23
PET	22-45	0.214	0.15	24
Silk/PLA/CNT yarn	22-45	0.268	24	This work

1. Shi, S. et al. A Bionic Skin for Health Management: Excellent Breathability, In Situ Sensing, and Big Data Analysis. *Adv. Mater.* **36**, 2306435 (2024).
2. Liang, X. et al. Thermal Transfer Printed Flexible and Wearable Bionic Skin with Bilayer Nanofiber for Comfortable Multimodal Health Management. *Adv. Healthc. Mater.* **14**, 2403780 (2025).
3. Tian, H. et al. Hierarchical Fermat helix-structured electrochemical sensing fibers enable sweat capture and multi-biomarker monitoring. *Mater. Horiz.* **10**, 5192-5201, (2023).
4. Mei, X., Yang, J., Liu, J. & Li, Y. Wearable, nanofiber-based microfluidic systems with integrated electrochemical and colorimetric sensing arrays for multiplex sweat analysis. *Chem. Eng. J.* **454**, 140248 (2023).
5. Wang, L. et al. Weaving Sensing Fibers into Electrochemical Fabric for Real-Time Health Monitoring. *Adv. Funct. Mater.* **28**, 1804456 (2018).
6. Salvo, P. et al. Temperature and pH sensors based on graphenic materials. *Biosens. Bioelectron.* **91**, 870-877 (2017).
7. Sun, Y. et al. Stretchable and Smart Wettable Sensing Patch with Guided Liquid Flow for Multiplexed in Situ Perspiration Analysis. *ACS Nano* **18**, 2335-2345 (2024).
8. Ji, W. et al. Large-scale fully printed “Lego Bricks” type wearable sweat sensor for

- physical activity monitoring. *NPJ Flex. Electron.* **7**, 53 (2023).
9. An, Z. et al. Body Heat Powered Wirelessly Wearable System for Real-time Physiological and Biochemical Monitoring. *Adv. Funct. Mater.* **33**, 2303361 (2023).
 10. He, W. et al. Integrated textile sensor patch for real-time and multiplex sweat analysis. *Sci. Adv.* **5**, eaax0649 (2019).
 11. Niu, J. et al. A Fully Elastic Wearable Electrochemical Sweat Detection System of Tree-Bionic Microfluidic Structure for Real-Time Monitoring. *Small* **20**, 2306769 (2024).
 12. Gai, Y. et al. A Self-Powered Wearable Sensor for Continuous Wireless Sweat Monitoring. *Small Methods* **6**, 2200653 (2022).
 13. Tong, X. et al. Multifunctional Fiber for Synchronous Bio-Sensing and Power Supply in Sweat Environment. *Adv. Funct. Mater.* **33**, 2301174 (2023).
 14. Mo, L., Ma, X., Fan, L., Xin, J. H. & Yu, H. Weavable, large-scaled, rapid response, long-term stable electrochemical fabric sensor integrated into clothing for monitoring potassium ions in sweat. *Chem. Eng. J.* **454**, 140473 (2023).
 15. Nyein, H. Y. Y. et al. A Wearable Electrochemical Platform for Noninvasive Simultaneous Monitoring of Ca^{2+} and pH. *ACS Nano* **10**, 7216-7224 (2016).
 16. Cai, X. et al. Fully Integrated Multiplexed Wristwatch for Real-Time Monitoring of Electrolyte Ions in Sweat. *ACS Nano* **18**, 12808-12819 (2024).
 17. Zareh, M. M., Mohamed, S. F. & Elsheikh, A. M. Polymeric Electrochemical Sensor for Calcium Based on DNA. *Polymers* **14**, 1896 (2022).
 18. Soleimani, A. et al. Towards sustainable and humane dairy farming: A low-cost electrochemical sensor for on-site diagnosis of milk fever. *Biosens. Bioelectron.* **259**, 116321 (2024).
 19. Kuzubasoglu, B. A., Sayar, E., Cochrane, C., Koncar, V. & Bahadir, S. K. Wearable temperature sensor for human body temperature detection. *J. Mater. Sci-Mater. El.* **32**, 4784-4797 (2021).
 20. Khalaf, A. M., Issa, H. H., Ramirez, J. L. & Mohamed, S. A. All Inkjet-Printed Temperature Sensors Based on PEDOT: PSS. *IEEE Access* **10**, 61094-61100 (2022).
 21. Dankoco, M. D., Tesfay, G. Y., Benevent, E. & Bendahan, M. Temperature sensor realized by inkjet printing process on flexible substrate. *MSEB* **205**, 1-5 (2016).
 22. Kuzubasoglu, B. A., Sayar, E. & Bahadir, S. K. Inkjet-Printed CNT/PEDOT:PSS Temperature Sensor on a Textile Substrate for Wearable Intelligent Systems. *IEEE Sens. J.* **21**, 13090-13097 (2021).

23. Zhang, Y. et al. High-linearity graphene-based temperature sensor fabricated by laser writing. *J. Mater. Sci-Mater. El.* **35**, 109 (2024).

24. Ma, S. et al. Ultra-Sensitive and Stable Multiplexed Biosensors Array in Fully Printed and Integrated Platforms for Reliable Perspiration Analysis. *Adv. Mater.* **36**, 2311106 (2024).

Revision:

(Page 15 in the revised manuscript)

“A comparison of the sensitivity between as-spun biosensors and previously reported patch^{22,24,25,41-53} or textile biosensors^{21,54-59} was presented in Fig. 3i and Supplementary Fig. 17. Furthermore, the detection range and long-term stability were discussed in Supplementary Tables 1-5. The comparisons highlighted the high sensitivity and stability of as-spun biosensors within the human physiological range, surpassing that of the majority of existing wearable biosensors in the field.”

2. The distribution statistics of the spun yarn sensor pore structure (200-500 nm) are missing. It is suggested to supplement the aperture distribution histogram.

Response: Thank you for the valuable remark. We have supplemented the BJH pore size distribution curve of the CSCP yarn obtained through the N₂ adsorption-desorption technique, as shown in new Supplementary Fig. 11b. The pore size distribution analysis revealed a predominant concentration of nanopores within the 200-500 nm range. These structural characteristics demonstrated that the as-spun CSCP yarn possessed an interconnected nanoporous network with elevated specific surface area, facilitating efficient sweat absorption and transportation.

In response to the reviewer’s insightful suggestion, we have added new Supplementary Fig. 11b in the revised supplementary information, along with a related description in the revised manuscript (highlighted in blue).

Supplementary Fig. 11b Distribution curve of CSCP yarn (dV/dD—pore diameter curve).

Revision:

(Page 11 in the revised manuscript)

“Both SEM characterization and pore size distribution analysis revealed that the CSCP yarn featured a porous structure with pore sizes ranging from 200 to 500 nm (Fig. 2g and Supplementary Fig. 11b). This hierarchical porosity not only facilitated efficient sweat absorption but also provided numerous reactive sites for ion and electron transfer.”

3. The article mentions that "interfering ions cause potential fluctuations of less than 1 mV", but does not specify the specific concentration range of interfering matter. The concentration gradient data of the interference experiment should be supplemented to verify the selectivity.

Response: Thank you for the constructive remark. For the anti-interference experiment, we have included data for an interfering ion concentration of 1 mM in the revised manuscript. Additionally, we have completed anti-interference experiments using interfering ion concentrations of 5 mM and 10 mM (new Supplementary Fig. 15). The as-spun pH, Na⁺, K⁺, and Ca²⁺ biosensors all exhibited high sensitivity to the target molecule, and can respond within 1 second. Regardless of the concentration, the as-spun biosensors exhibited a potential fluctuation of less than 1 mV with the adding of interfering ions, demonstrating their excellent anti-interference capability and the ability to disregard the impact of common ions on their potential.

In response to the reviewer’s insightful suggestion, we have added new Supplementary Fig. 15 in the revised supplementary information, along with a related description in the revised manuscript (highlighted in blue).

Supplementary Fig. 15 Selectivity of the as-spun biosensors. a-h Selectivity with the interfering materials in 5 mM (a-d) and 10 mM (e-f).

Revision:

(Page 14 in the revised manuscript)

“Notably, as-spun biosensors exhibited a potential fluctuation of less than 1 mV in response to interfering ions of varying concentrations, demonstrating their excellent ability to disregard the impact of common ions (Fig. 3d-g and Supplementary Fig. 15).”

4. Although the article mentions that the sensor remains stable within 5000 seconds, practical applications require consideration of continuous use performance over longer periods. It is recommended to supplement long-term stability test data (e.g. 24 hours of continuous monitoring results).

Response: Thank you for the constructive remark. We agree that long-term stability is essential for the validation process of biosensors. With the reviewer’s suggestion, we have conducted additional experiments to substantiate the stability of our device.

The long-term stability of the multi-biosensors was evaluated by continuously testing in standard solutions for 24 hours. The measured potentials and resistance demonstrated remarkable stability across all as-spun biosensors. Specifically, the as-spun biosensors exhibited a drift of 0.13, 0.17, 0.1, 0.19 mV h⁻¹ for pH, Na⁺, K⁺, Ca²⁺ sensing, and 0.05 Ω h⁻¹ for temperature sensing. Therefore, the remarkable long-term stability of the biosensors can fully meet the long-term requirements for practical applications.

We have modified Fig. 3c and Supplementary Fig. 14, along with a related discussion in the revised manuscript (highlighted in blue).

Fig. 3c Long-term stability of pH, Na⁺, K⁺ and Ca²⁺ biosensors in standard solutions.

Supplementary Fig. 14 Long-term stability of as-spun temperature sensor over 24 hours.

Revision:

(Page 14 in the revised manuscript)

“The durability of as-spun biosensors was also demonstrated through continuous operation in standard solutions for over 24 hours (Fig. 3c and Supplementary Fig. 14). The measured potentials and resistances demonstrated remarkable stability across as-spun biosensors. Specifically, as-spun yarns exhibited a drift of 0.13, 0.17, 0.1, 0.19 mV h⁻¹ for pH, Na⁺, K⁺, Ca²⁺ sensing, and 0.05 Ω h⁻¹ for temperature sensing. Therefore, the remarkable long-term stability of as-spun biosensors can fully meet the requirements for practical applications in the long-term health monitoring environment.”

5. In the biocompatibility experiment (Fig. 4c-d), only "n=4 independent experiments" is mentioned, but statistical methods of data processing are not explained. It is suggested to supplement statistical analysis to ensure the reliability of experimental results.

Response: Thank you for the remark. We have utilized the box diagram to conduct a statistical analysis of the activity of cells co-cultured with different as-spun biosensors in the CCK-8 experiment, as shown in new Supplementary Fig. 18.

The absorption of all cell groups increased significantly with prolonged culture time, indicating that the ionic environment has a continuous impact on cell activity, possibly due to enhanced cell proliferation. Of these, there was a significant increase in absorbance corresponding to the Na⁺ and Ca²⁺ biosensors, whereas the increase in pH and K⁺ biosensors was relatively flat. The strong pro-cell activity effects of Na⁺ and Ca²⁺ implied a key role in cell signaling or energy metabolism, while the stable low values of the pH group may reflect its strict dependence on cell homeostasis. Univariate analysis of variance showed that the difference of absorbance among different ion groups was statistically significant at 24 hours (P < 0.01), 48 hours (P < 0.01) and 72 hours (P < 0.01). The standard deviation of experimental data was less than 0.05, which indicated that the results were highly repeatable and reliable. Thus, the results confirmed the time-dependent regulation of cell activity by biosensors, of which the most significant contribution was given by Na⁺ biosensors and Ca²⁺ biosensors, while the effect of pH biosensors was relatively conservative.

With the reviewer's comment, we have added new Supplementary Fig. 18 in the revised supplementary information, along with a related description in the revised manuscript (highlighted in blue).

Supplementary Fig. 18 Box-and-whisker plot of mouse epithelial cells cultured in biocompatibility test. a-c CCK-8 experimental results within 24 h (a), 48 h (b) and (c) 72h. The dashed lines indicate the upper and lower limits of the absorbance for the cells. The box ends represent the 25th and 75th percentiles. The horizontal line in each box represents the median. The upper and lower whiskers represent the maxima and minima, respectively, which refer to the range of non-outlier data values.

Revision:

(Page 17 in the revised manuscript)

“Univariate analysis of variance showed that the difference of absorbance among different ion groups was statistically significant at 24 hours (P < 0.01), 48 hours (P <

0.01) and 72 hours ($P < 0.01$). The standard deviation of the experimental data was less than 0.05, which indicated that the results were highly repeatable and reliable (Supplementary Fig. 18).”

6. The performance of biosensors also needs to be tested in extreme environments such as high temperature and humidity.

Response: Thank you for the constructive suggestion. We agree that the stability of biosensors under extreme environments is essential for the practical application of wearable biosensors. With the suggestion, we have conducted additional experiments to substantiate the stability of our as-spun biosensors.

The as-spun biosensors underwent systematic environmental stability testing using a constant temperature heating table (set at 25°C, 45°C, 65°C) in conjunction with a custom-built humidity box (maintained at 40%, 60%, 80% RH). During this process, standardized analyze solutions (pH, Na⁺, K⁺, and Ca²⁺) were dropped on the biosensors. As shown in new Supplementary Fig. 19 below, the biosensors exhibited exceptional signal consistency under different temperature and humidity environments, with maximum voltage deviations constrained within ± 1.5 mV. These results demonstrated the excellent stability of the biosensor under high temperature and humidity conditions, thus confirming its suitability for practical applications involving environmental extremes.

With the reviewer’s insightful suggestion, we have added new Supplementary Fig. 19 in the revised supplementary information, along with a related description in the revised manuscript (highlighted in blue).

Supplementary Fig. 19 Stability of as-spun biosensors against environmental factors. a, b Stability of the biosensors evaluated at different temperatures in a constant temperature heating table. c, d Stability of the biosensors evaluated under varying humidity conditions in a custom-built humidity box.

Revision:

(Page 17 in the revised manuscript)

“The stability of as-spun biosensors was evaluated under high temperature and humidity conditions using a constant temperature heating table and a custom-built humidity box. During testing, standardized analyze solutions were applied to the biosensors. The multi-biosensor array demonstrated exceptional signal consistency across different environmental conditions, with maximum voltage deviations constrained within ± 1.5 mV (Supplementary Fig. 19).”

7. Some terms are used inconsistently (e.g., "sweat acquiring" vs. "sweat capture"), and there are a few syntax errors. It is suggested to unify the terminology and polish the language throughout the article.

Response: Thank you for the constructive remark. “Sweat acquiring” has been replaced with “sweat capture” throughout the manuscript, and all terms have been checked and harmonized. Additionally, a native English speaker has been engaged to polish the manuscript, ensuring that the revised version meets the language standards.

Reviewer #2

Comments: The article by Ming Li et al. discusses the development of a smart hairband designed for monitoring various biochemical and biophysical parameters. The research topic is highly relevant, and the results presented in the article are promising, although some aspects require clarification. However, the presentation of the results needs improvement to meet the standards necessary for publication in Nature Communications. In particular, the authors should emphasize the novelty of their work while maintaining a high scientific rigor throughout the manuscript. For this reason, the article is not publishable in its proposed format; it could only achieve the required level for publication after significant revisions.

Response: We appreciate the reviewer's interest in our studies of as-spun biosensor array for health monitoring applications, and thanks for the encouragement and valuable remarks. A systematic overview of the technical advancements in wearable and textile-based electrochemical biosensors has been provided. The fabrication and structural characterization of the as-spun biosensors are also described and discussed in detail. We have made the requested revisions, and all the details of our modifications are indicated in our responses and the corresponding modifications. All changes are highlighted in the revised version of the manuscript.

Main comments:

1) The introduction should provide background on the recent literature regarding wearable sensors, with a particular focus on textile sensors. While the authors effectively explain the general motivations behind the development of these sensors, they do not discuss the technical advancements necessary to achieve the Technology Readiness Level required for commercialization. What type of transduction is primarily used in the design of wearable sensors, and why? What materials are predominantly utilized in the fabrication of wearable sensors? What configurations are employed in the creation of wearable and textile-based electrochemical sensors? It is essential to mention amperometric, potentiometric, and transistor-based designs. A comparison of the results with recent literature would help better contextualize the device's performance. What innovative aspects does this research present compared to the current state of the art? All these points should be thoroughly discussed. Additionally, it would be beneficial to cite the following works: 10.1038/nature16521 and <https://doi.org/10.1038/s41598-020-74337-w>. All this information must be integrated

into the text.

Response: Thank you for the constructive feedback. In response to your suggestion, we have reconstructed the introduction to provide a comprehensive overview of recent advancements in wearable and textile-based electrochemical biosensors. And a comparison of the results with recent literature is provided in Fig.3i, new Supplementary Fig. 17 and new Supplementary Tables 1-5. Key updates include:

1. Transduction type

Based on transduction mechanisms, electrochemical detection can be broadly categorized into six modalities: potentiometric, amperometric, voltammetric, organic electrochemical transistors (OECTs), photoelectrochemical (PEC), and electrochemiluminescence (ECL). Among these, potentiometric and amperometric biosensors are mainly used for wearable sensor design due to their simplified device architecture and high sensitivity.

2. Configuration

Wearable electrochemical sensors mainly consist of flexible substrate and functional layer, relying on a dual-electrode system for biometric recognition.

3. Materials

Flexible polymers, hydrogels, and textiles have been widely adopted as substrate materials for wearable sensors, with textile-based architectures standing out due to their intrinsic breathability, capillary-driven fluid transport mechanisms, and natural compatibility with conventional apparel systems.

4. Innovation

The CSCP biosensor pioneers a monolithic biosensing fiber forming strategy that fundamentally differs from conventional layer-by-layer assembly methods.

5. Comprehensive comparison

A comprehensive comparison of the performance between our as-spun biosensors and those reported in recent literature is provided in Fig.3i, new Supplementary Fig. 17 and Supplementary Tables 1-5.

Fig. 3i Sensitivity comparison of the as-spun multimodal biosensors with other reported

patch or fiber biosensors.

Supplementary Fig. 17 Sensitivity comparison of as-spun temperature sensor with other reported patch or fiber sensors.

The introduction has been restructured in the revised manuscript, and new Supplementary Fig. 17 and new Supplementary Tables 1-5 have been added to the Supplementary Information. The references of 10.1038/nature16521 and 10.1038/s41598-020-74337-w provide valuable insights for this work, which have been cited as Ref. 20 and Ref. 26 in the revised manuscript. All substantive modifications, including textual revisions and supplemental material additions, have been explicitly highlighted in blue throughout the revised manuscript.

Revision:

(Page 2-4 in the revised manuscript)

“Blood analysis remains an indispensable tool in medical monitoring, providing critical physiological and pathological information to support disease diagnosis, therapeutic evaluation, and dynamic health monitoring^{1,2}. However, the reliance on invasive sampling and centralized laboratory infrastructure significantly restricts real-time monitoring capabilities, particularly in emergencies requiring immediate clinical intervention. To overcome these constraints, extensive research has focused on developing real-time, autonomous biomarker analysis systems utilizing non-invasive biofluids^{3,4}. Among alternative biofluids, sweat has emerged as a particularly promising medium due to the ubiquitous distribution of sweat glands across the human body^{5,6}. Additionally, sweat is rich in biomarkers, including electrolyte, metabolite, nutrition, and hormone, which reflect physiological conditions at the molecular level and track health status⁷⁻⁹. Recent studies have demonstrated clinically significant correlations between sweat biomarkers concentrations and corresponding blood analyte levels¹⁰⁻¹², providing a scientific foundation for sweat-based diagnostics. Although sweat offers

unique advantages for real-time health monitoring, significant challenges remain in practical implementation. Firstly, dynamic compositional fluctuations and ultralow biomarker concentrations necessitate exceptional sensitivity and ultra-low detection limit^{13,14}. Secondly, reliance on passive secretion or external stimulation limits continuous acquisition of samples^{15,16}. These intrinsic constraints have motivated continuous advancements in sweat analysis system for active biofluid harvesting and precise biomarker quantification.

Electrochemical biosensors dominate sweat analysis owing to their high sensitivity, selectivity, and miniaturization potential^{17,18}. Based on transduction mechanisms, electrochemical detection can be broadly categorized into six modalities¹⁹: potentiometric, amperometric, voltammetric, organic electrochemical transistors (OECTs), photoelectrochemical (PEC), and electrochemiluminescence (ECL). Potentiometric configurations are prevalent for ion-selective detection due to operational simplicity, while amperometric approaches excel in metabolite quantification through redox reactions. As an early innovation, Gao et al.²⁰ pioneered the first fully integrated wearable biosensor array on poly(ethylene terephthalate) (PET), combining potentiometric and amperometric modalities for multiplexed sweat analysis. Subsequent developments have also employed flexible polymeric substrates, such as polydimethylsiloxane (PDMS)²¹⁻²³ and polyimide (PI)^{24,25}, to enhance mechanical compliance with epidermal surfaces. However, the intrinsic hydrophobicity of polymer impedes interfacial fluid transport, thereby undermining continuous biosensing reliability. While microfluidic integration enhances biofluid collection, intricate fabrication and high cost remain unresolved issues. Additionally, poor breathability frequently induces skin irritation during prolonged wear, presenting critical barriers the long-term application of wearable biosensors.

Textiles with hierarchical fiber architectures spanning micro-to-nano scales provide distinct advantages for wearable integration, particularly in breathability, lightweight, and intrinsic compatibility with garment systems²⁶. While conventional textile biosensor fabrication predominantly employs drop-coating or printing techniques for functional layer deposition, these methods often exhibit structural inhomogeneity and performance variability²⁷. The deficiencies are attributable to weak interfacial binding energy between textile substrate and functional layer, resulting in progressive delamination and performance degradation²⁸. Such interfacial failures represent fundamental limitations of additive manufacturing strategies based on post-fabrication

functionalization. In contrast, wet spinning is a monolithic yarn formation technique, whereby spinning dope is extruded into a coagulation bath and further utilized solvent-nonsolvent dual diffusion to induce phase separation²⁹⁻³¹. Incorporating biosensing elements into spinning dope enables the fabrication of monolithic sensing yarns with seamless conductive-sensing networks, simultaneously achieving shortened electron transfer pathways and directional sweat transport^{32,33}. Furthermore, the comfortable yarn structure mitigates allergic or inflammatory reactions during direct skin contact with textile-based biosensors^{34,35}. Such systems mark a significant advancement in wearable diagnostics by eliminating interfacial delamination risks, thereby establishing a robust platform for emergency healthcare monitoring through reliable wearable sensor systems.”

20. Gao, W. et al. Fully integrated wearable sensor arrays for multiplexed in situ perspiration analysis. *Nature* **529**, 509-514 (2016).

26 Possanzini, L. et al. Textile sensors platform for the selective and simultaneous detection of chloride ion and pH in sweat. *Sci. Rep.* **10**, 17180 (2020).

(Page 15 in the revised manuscript)

“. A comparison of the sensitivity between as-spun biosensors and previously reported patch^{22,24,25,41-53} or textile biosensors^{21,54-59} was presented in Fig. 3i and Supplementary Fig. 17. Furthermore, the detection range and long-term stability were discussed in Supplementary Tables 1-5. The comparisons highlighted the high sensitivity and stability of as-spun biosensors within the human physiological range, surpassing that of the majority of existing wearable biosensors in the field.”

2) The structure of the wearable sensor needs to be described and discussed in greater detail, particularly regarding the reference electrode. In the wearable sensor, the reference electrode is a strip of Ag/AgCl, whereas in sensor characterization, a traditional electrode is used. The potential of Ag/AgCl depends on the Cl⁻ concentration of the solution in which it is immersed, making it crucial to study potential interferences with the final device. The authors should provide a more comprehensive description of the device's structure while identifying how the presence of Cl⁻ interferes with measurements. All interference tests must be conducted using the final device, and the interference from chlorides should be discussed in detail.

Response: We thank the reviewer for this suggestion. We apologize for the unclear description of the electrochemical measurements in the manuscript. All sensor tests, including sensor characterization and wearable applications, are performed in the two-electrode system with CSCP-based biosensor as the working electrode, while an Ag/AgCl yarn serves as both the reference and counter electrodes.

The schematic diagram and cross-section scanning electron microscopy (SEM) images of SCP-based biosensor, CSCP-based biosensor, and Ag/AgCl yarn have been supplemented as shown in new Supplementary Fig. 7 and 9 below. The SCP-based biosensor was spun by extruding functional SCP mixture into coagulation bath, resulting in a homogeneous yarn architecture with controlled diameter. In contrast, the CSCP-based biosensor was fabricated through the coaxial wet spinning strategy. The custom-designed coaxial needle features two isolated microchannels: a central channel for continuous silk yarn feeding and a concentric annular channel for concurrent deposition of functional SCP sheath. Schematic diagrams and cross-sectional SEM images revealed that both the SCP and CSCP based biosensors exhibited exceptional structural stability, morphological consistency, and compositional homogeneity. Specifically, the CSCP-based biosensor possessed an optimized core-sheath structure with a central silk yarn, which achieve a 17-fold improvement in mechanical strength.

Building on this, the Ag/AgCl yarn was fabricated by first coating Ag/AgCl conductive slurry onto the CSCP yarn and then encapsulated with a layer of PVB solution containing 79.1 mg PVB and 50 mg NaCl in 1 mL of methanol. Microstructural characterizations revealed that the Ag/AgCl conductive layer uniformly coated the CSCP yarn surface, thus forming a stable reference electrode interface. The NaCl/PVB membrane effectively fixed the reference yarn potential by maintaining Cl^- activity and preventing external Cl^- interference. The potential of the reference yarn was found to be well-maintained under varying salt concentrations (new Supplementary Fig. 10).

We have added new Supplementary Fig. 7, 9, and 10 in the revised supplementary information, along with a related description in the revised manuscript (highlighted in blue).

Supplementary Fig. 7 Schematic illustration and structural characterization of as-spun biosensors. a, b Schematic diagrams of SCP-based biosensor (a) and CSCP biosensor (b). c, d Cross-section SEM images of SCP-based biosensor (c) and CSCP biosensor (d). Scale bars, 100 μm .

Supplementary Fig. 9 Schematic illustration and structural characterization of Ag/AgCl yarn. a Schematic diagram of Ag/AgCl yarn. b Cross-section SEM image of Ag/AgCl yarn. Scale bars, 100 μm . c EDS image of Ag/AgCl yarn.

Supplementary Fig. 10 Stability of the Ag/AgCl yarn in PBS solutions containing 5 mM

and 10 mM of different saline solution.

Revision:

(Page 9 in the revised manuscript)

“Schematic diagrams and cross-sectional SEM images revealed that both SCP and CSCP based biosensors exhibited exceptional structural stability, morphological consistency, and compositional homogeneity. Specifically, the CSCP-based biosensor possessed an optimized core-sheath structure with a central silk yarn, which may provide a certain mechanical support (Supplementary Fig. 7).”

(Page 11 in the revised manuscript)

“Based on this, the reference electrode was fabricated by first coating Ag/AgCl conductive slurry onto the 6% CNT-incorporated CSCP yarn and then encapsulating it with the NaCl-saturated PVB membrane. Microstructural analysis confirmed that a continuous Ag/AgCl layer was uniformly adhering to the yarn surface, forming a stable reference electrode interface (Supplementary Fig. 9). The NaCl/PVB membrane effectively fixed the reference yarn potential by maintaining Cl^- activity and preventing external Cl^- interference. The potential of the Ag/AgCl yarn was found to be well-maintained under varying salt concentrations (Supplementary Fig. 10).”

(Page 25 in the revised manuscript)

“followed by the encapsulated with a layer of PVB solution containing 79.1 mg PVB and 50 mg NaCl in 1 mL of methanol.”

(Page 26 in the revised manuscript)

“while an Ag/AgCl yarn served as the reference and counter electrodes.”

3) Fiber characterization and fabrication. Why do the authors state that the fiber are core-sheath structured? The authors should provide experimental evidence of it. The text is not clear about the difference between CSCP and SCP. This difference should be discussed describing the fabrication procedures and discussing the experimental results. Finally the electrochemical characterization should be enlarged in the main text.

Response: Thank you for the valuable remark. Wet spinning is a fiber production technique involving the continuous extrusion of polymer solutions into a coagulation

bath to form filaments. The fundamental principle is the utilization of solvent-nonsolvent dual diffusion to induce polymer phase separation, followed by controlled solidification into nascent fibers.

The fabrication of SCP and CSCP based biosensors both employed a self-assembled wet spinning system, wherein a 16 wt% functional SCP mixture was extruded into channels with deionized water serving as the coagulation bath. The phase separation-induced solidification process, driven by dual-diffusion between 1,4-dioxane solvent and aqueous coagulant, enabled continuous production of monolithic sensing yarn through roller collection (new Supplementary Fig. 6a). The main difference between SCP and CSCP based biosensors arises from their respective needle, which dictates their final architectures. Specifically, the SCP-based biosensor was fabricated through a conventional single-channel spinning needle, enabling direct extrusion of the functional SCP mixture into the coagulation bath to form homogeneous yarn (Supplementary Fig. 6b, c). In contrast, the CSCP-based biosensor utilized a coaxial needle with dual isolated microchannels—a central channel for continuous silk yarn feeding and a concentric annular channel for concurrent deposition of functional SCP sheath. Subsequent solvent-nonsolvent dual-diffusion facilitated the formation of a monolithic sensing yarn with core-sheath structure (Supplementary Fig. 6d, e).

Schematic diagrams and cross-sectional SEM images revealed that both the SCP and CSCP based biosensors exhibited exceptional structural stability, morphological consistency, and compositional homogeneity. Specifically, the CSCP-based biosensor possessed an optimized core-sheath structure with a central silk yarn core circumferentially sheathed by the functional SCP mixture (Supplementary Fig. 7). The structural design is attributed to the cohesive properties of the viscous spinning dope, thus providing superior resistance to structural failure or interfacial delamination, maintaining structural continuity during operational stresses.

We have added new Supplementary Fig. 6 and 7 in the revised supplementary information, along with a related description in the revised manuscript (highlighted in blue). Furthermore, the electrochemical characterization has been enlarged in the main text (highlighted in blue).

Supplementary Fig. 6 Fabrication of as-spun biosensors through wet spinning technique. a Continuous preparation of the biosensing yarn through self-assembled wet-spinning device. b, c Optical image and diagrammatic sketch of SCP-based biosensor preparation through conventional wet-spinning production. d, e Optical image and schematic diagrams of CSCP-based biosensor preparation through coaxial wet-spinning production.

Supplementary Fig. 7 Schematic illustration and structural characterization of as-spun biosensors.

biosensors. a, b Schematic diagrams of SCP-based biosensor (a) and CSCP biosensor (b). c, d Cross-section SEM images of SCP-based biosensor (c) and CSCP biosensor (d). Scale bars, 100 μm .

Revision:

(Page 9 in the revised manuscript)

“For the SCP-based biosensor fabrication, a single-channel spinning needle was employed to induce direct phase inversion of the functional SCP mixture, resulting in a homogeneous yarn architecture with controlled diameter. In contrast, the core-sheath SCP (CSCP)-based biosensor utilized a coaxial needle with dual isolated microchannels—a central channel for continuous silk yarn feeding and a concentric annular channel for concurrent deposition of functional SCP mixture. Schematic diagrams and cross-sectional SEM images revealed that both SCP and CSCP based biosensors exhibited exceptional structural stability, morphological consistency, and compositional homogeneity. Specifically, the CSCP-based biosensor possessed an optimized core-sheath structure with a central silk yarn, which may provide a certain mechanical support (Supplementary Fig. 7).”

(Page 10 in the revised manuscript)

“With regard to electrochemical characteristic, CSCP yarns with varying CNT ratios (2-8%) were analyzed in $\text{K}_3\text{Fe}(\text{CN})_6/\text{KCl}$ solution to determine the optimal CNT ratio, thus achieving minimized charge transfer resistance and maximized electrochemical activity. Electrochemical impedance spectroscopy demonstrated that the 6% CNT-incorporated CSCP yarn exhibited a significantly reduced semicircle radius, exhibiting minimal charge-transfer resistance among all tested yarns (Supplementary Fig. 8a). Complementary cyclic voltammetry revealed the highest Faraday current⁴¹ and improved capacitive behavior in 6% CNT-incorporated CSCP yarn, suggesting optimized interfacial electron transfer kinetics and surface redox activity. Furthermore, the redox coupling indicated its superior charge storage capacity and electrochemical activity (Supplementary Fig. 8b). These characteristics demonstrated exceptional and stable electrochemical performance of 6% CNT-incorporated CSCP yarn in simulated sweat environment, confirming its suitability for biosensing applications.”

(Page 24-25 in the revised manuscript)

“The as-spun biosensors were prepared through a self-assembled wet-spinning system, wherein the 16 wt% functional SCP mixture was loaded into a syringe pump and extruded via a needle into a coagulation bath composed of deionized water. Specifically, SCP-based biosensor fabrication utilizes a conventional single-lumen needle (19 G), while CSCP-based biosensor production incorporates coaxial needle with silk yarn acting as the core (14/19 G).”

4) Analytical concerns of proposed sensors. The directional transport of sweat is a key point of proposed device because it is the sampling system that guarantee the continuous monitoring of the human perspiration. The authors discuss this point at lines 102-109. A more detailed discussion (and more characterizations) must be added in the text in order to provide the reader the information about this important point of the proposed device. The compounds (also the analytes) that are present on the skin wearer can interfere with the measurement. How does the authors avoid this problem? The authors describe the operating principles of the sensors in lines 194 – 197. This discussion should be followed the literature concerning potentiometric sensors. The sensors does not used ion channel (no membrane protein has been added to the sensors). The potential change is proportional to the concentration (activity) logarithm and not to the biding strength, as demonstrated by the calibration plots.

Response: Thank you for the valuable remark. The wettability behavior of a liquid on the solid surface is governed by the solid-liquid-gas three-phase boundary, which can be described by Young's model, Wenzel model and Cassie-Baxter model. For an ideally smooth solid surface, Young's equation can be obtained by balancing the horizontal surface tension in the system (Eq. (1)). However, solid surfaces exhibit varying degrees of roughness in reality. In such scenarios, the contact angles predicted by the Wenzel model and the Cassie-Baxter model are expressed by Eq. (2) and (3), respectively.

$$\cos\theta_Y = \frac{\gamma_{sg} - \gamma_{sl}}{\gamma_{lg}} \quad (1)$$

$$\cos\theta_w = r \frac{\gamma_{sg} - \gamma_{sl}}{\gamma_{gl}} \quad (2)$$

$$\cos\theta_{CB} = f(\cos\theta_Y + 1) - 1 \quad (3)$$

where θ_Y , θ_w and θ_{CB} are the equilibrium contact angle of the solid surface of Young's, Wenzel and Cassie-Baxter, respectively, γ_{sg} , γ_{sl} and γ_{lg} are the surface tension of solid-gas, solid-liquid and liquid-gas, respectively, r is the roughness coefficient, and f is the ratio of the contact area of the solid-liquid interface to the actual surface area within a

unit area.

Based on Eq. (1)-(3), it is evident that roughness significantly impacts the hydrophilic properties of a surface. Therefore, gradient wettability surfaces can be formed by employing asymmetric contact angles to induce a micro-roughness gradient. Driven by the surface energy difference, droplets are propelled along the wettability gradient, facilitating their coalescence and subsequent merging into larger droplets. This approach allows for precise point-to-point fluid control and regulation (Figure 2e). The interfacial imbalance force, commonly referred to as the wetting gradient driving force, can be mathematically expressed as Eq. (4):

$$F = 2R\gamma_{lg} \int_0^{\pi/2} (\cos\theta_A - \cos\theta_B) \cos\phi d\phi \quad (4)$$

where R is the base radius of the contact between the droplet and the interface, θ_A and θ_B respectively represent the contact angle of the droplet before and after the wetting gradient, and ϕ represents the polar angle.

In our work, the as-spun biosensors exhibit rapid water absorption and storage capabilities, which are attributed to its uniform microporous architecture. The contact angle was measured to be 61° when a liquid drop dropped on the yarn, with complete droplet absorption occurring within 30 seconds, demonstrating superior hydrophilic properties. Conversely, the textile substrate composed of hydrophobically modified yarns displayed evident non-wetting properties, maintaining incomplete droplet absorption even after 120 seconds of exposure (Supplementary Fig. 3 and Supplementary Fig. 4). Therefore, the integrated hairband constructed by as-spun biosensors and textile substrates exhibited significant hydrophilic and hydrophobic differences. This marked contrast in surface wettability created a directional liquid transportation in the integrated hairband, which could be explained by Young's model, Wenzel model and Cassie-Baxter model (Supplementary Fig. 5a). A wettability gradient was established at the interface between as-spun biosensors and textile substrates, as a result of an asymmetric contact angle distribution. This gradient can induce spontaneous fluid migration driven by surface energy differentials, thereby enabling precise fluidic capture and regulation of the as-spun biosensors (Supplementary Fig. 5b).

Supplementary Fig. 4 Wetting property of as-spun biosensor and textile substrate. a, b Diffusing process of the droplet on as-spun biosensor (a) and hydrophobic treated textile substrate (b).

Supplementary Fig. 5 Schematic illustration of sweat directional transport to multi-biosensor array. a Wettability behavior of a liquid on the solid surface. b Sweat capture ability of multi-biosensor array.

A systematic investigation is conducted to evaluate the interference effects of sweat electrolytes/metabolites, sebum secretions, and stratum corneum metabolites on the multi-biosensing hairband. The previous experiments revealed that as-spun biosensors exhibited potential fluctuations below 1 mV when exposed to sweat electrolytes/metabolites, confirming their superior anti-interference capability (Fig. 3d-g). The integrated multi-biosensing hairband also demonstrated stable performance in long-term wearable applications. Its performance was unaffected by sebum secretions and stratum corneum metabolites, which can be effectively removed through regular washing. Due to the intrinsic textile structural characteristics and mechanical robustness of as-spun biosensors, the integrated multi-biosensing hairband demonstrated exceptional laundering stability, retaining over 97% initial potential or

resistance after 30 laundering cycles (Supplementary Fig. 20). This approach effectively removes sebum secretions, stratum corneum metabolites, and sweat-derived crystal deposits from the as-spun biosensors, thereby maintaining the sensing performance.

Supplementary Fig. 20 Stability of as-spun biosensors under regular washing.

With the reviewer's suggestion, we have added new Supplementary Fig. 4, 5 and 20 in the revised supplementary, along with a related description in the revised manuscript (highlighted in blue). In addition, the operating principles of the biosensors have been revised in the main text (highlighted in blue).

Revision:

(Page 6-7 in the revised manuscript)

“The as-spun biosensors demonstrated rapid water absorption and storage capabilities, attributable to their uniform microporous architecture. The contact angle was measured to be 61° , with complete droplet absorption occurring within 30 seconds, demonstrating superior hydrophilic properties. Conversely, the textile substrate composed of hydrophobically modified yarns displayed evident non-wetting properties, maintaining incomplete droplet absorption even after 120 seconds (Supplementary Fig. 3 and Supplementary Fig. 4). According to the surface wettability theory, the asymmetric contact angle distribution established a wettability gradient between the biosensing yarns and the textile substrate (Supplementary Fig. 5). The gradient facilitated the directional transport of sweat, thereby enabling precise fluidic capture and regulation of the biosensor array (Fig. 1d).”

(Page 13 in the revised manuscript)

“Ion biosensors utilize the functionalized SCP mixture to selectively bind to target ions, leading to the potential gradient at the interface. The generated electromotive force

adheres to the Nernstian equation, allowing for the precise characterization of ion concentration levels in sweat.”

(Page 17-18 in the revised manuscript)

“Leveraging the intrinsic textile structure and mechanical robustness of as-spun biosensors, the multi-biosensing hairband demonstrated exceptional laundering stability, retaining over 97% initial potential or resistance after 30 laundering cycles (Supplementary Fig. 20). This approach effectively removed sebum secretions, stratum corneum metabolites, and sweat-derived crystalline deposits from as-spun biosensors, thereby maintaining the sensing performance.”

5) The title should be changed. Carrying out the experiments in an ambulance setting does not provide valuable evidence considering emergency use. Moreover, the sensors responses are close to the response of common potentiometric sensors. Therefore, the sensors are not ultrasensitive.

Response: We thank the reviewer for this suggestion. The choice of ambulance scenarios for the study is driven by the key limitations of the current emergency health assessment. Conventional blood diagnosis relies on centralized laboratory infrastructure and protracted analysis times, which poses a significant challenge in mobile emergency scenarios. The multi-biosensing hairband incorporates portability and real-time performance capabilities, which may prove an effective solution to the issue. Furthermore, the remarkable robustness of the multi-biosensors underscores the suitability for applications involving mechanical deformation and extreme environments. Considering that the reviewers might think that the ambulance scenario is single and limited, the multi-biosensing hairband can also be applied to extreme athletes, sick patients, elderly people, and military relief, etc (Supplementary Fig. 1).

Supplementary Fig. 1 Multi-scenario applications of multi-biosensing hairband for emergency health assessment.

With the reviewer's suggestion, we have changed the title and added Supplementary Fig. 1 in the revised manuscript.

Revision:

Title: Multi-Biosensing Hairband for Emergency Health Assessment

(Page 5-6 in the revised manuscript)

“In order to provide the real-time physiological information, a multi-biosensing hairband system with high sensitivity and rapid response has been developed for emergency health assessment (Fig. 1a and Supplementary Fig. 1).”

Other comments that require a text modification:

Abstract: abstract should be more quantitative and informative

Response: Thank you for the remark. We have revised the abstract to make it more quantitative and informative.

Revision:

(Page 1 in the revised manuscript)

“Blood analysis is regarded as the gold standard for monitoring analytes in clinical diagnostics. However, its time-consuming and site-limited nature often delays medical interventions that are crucial for patients in emergency scenarios. Herein, a novel multi-biosensor array is developed through coaxial wet spinning for real-time multiplex detection of sweat biomarkers (pH, Na⁺, K⁺, and Ca²⁺) and body temperature. The engineered microstructure of the multi-biosensor array exhibits a specific surface area approximately 200 times larger than that of conventional coated yarns, thereby facilitating directional sweat transport. The sensitivities of the pH, Na⁺, K⁺, Ca²⁺ and temperature sensors are determined to be 39.52 mV pH⁻¹ (3-7), 56.33 mV dec⁻¹ (10-160 mM), 34.13 mV dec⁻¹ (2-32 mM), 30.61 mV dec⁻¹ (0.5-2.53 mM), and 1.2 Ω °C⁻¹ (25-45 °C), respectively. It is noteworthy that the multi-biosensors exhibit consistent operational stability over 24 hours with minimal signal drift (pH: 0.13 mV h⁻¹, Na⁺: 0.17 mV h⁻¹, K⁺: 0.1 mV h⁻¹, Ca²⁺: 0.19 mV h⁻¹, temperature: 0.05 Ω h⁻¹). By integrating the multi-biosensors and a circuit patch into the textile substrate, a wireless hairband system is constructed for tracking human physiological dynamics. Such significant technological advancement offers an innovative strategy for constructing real-time

biosensing systems, which have the potential to revolutionize personalized healthcare and early diagnosis, particularly in emergency situations.”

Introduction. Sweat analysis. the weak point of sweat analysis should be discussed in the text.

Response: Thank you for the remark. We have supplemented the weak point of sweat analysis in the introduction part in the revised manuscript.

Revision:

(Page 2-3 in the revised manuscript)

“Although sweat offers unique advantages for real-time health monitoring, significant challenges remain in practical implementation. Firstly, dynamic compositional fluctuations and ultralow biomarker concentrations necessitate exceptional sensitivity and ultra-low detection limit^{13,14}. Secondly, reliance on passive secretion or external stimulation limits continuous acquisition of samples^{15,16}. These intrinsic constraints have motivated continuous advancements in sweat analysis system for active biofluid harvesting and precise biomarker quantification.”

13. Ye, C., Lukas, H., Wang, M., Lee, Y. & Gao, W. Nucleic acid-based wearable and implantable electrochemical sensors. *Chem. Soc. Rev.* **53**, 7960-7982 (2024).

14. Bariya, M., Nyein, H. Y. Y. & Javey, A. Wearable sweat sensors. *Nat. Electron.* **1**, 160-171 (2018).

15. Davis, N., Heikenfeld, J., Milla, C. & Javey, A. The challenges and promise of sweat sensing. *Nat. Biotechnol.* **42**, 860-871 (2024).

16. Yang, P., Wei, G., Liu, A., Huo, F. & Zhang, Z. A review of sampling, energy supply and intelligent monitoring for long-term sweat sensors *NPJ Flex. Electron.* **6**, 33 (2022).

Introduction and/or results: Spinning technology. Spinning technologies should be explained in order to provide the reader useful information about the fabrication technology.

Response: Thank you for the constructive suggestion. We have provided the explanation about spinning technology in the introduction part in the revised manuscript.

Revision:

(Page 4 in the revised manuscript)

“In contrast, wet spinning is a monolithic yarn formation technique, whereby spinning dope is extruded into a coagulation bath and further utilized solvent-nonsolvent dual diffusion to induce phase separation²⁹⁻³¹. Incorporating biosensing elements into spinning dope enables the fabrication of monolithic sensing yarns with seamless conductive-sensing networks, simultaneously achieving shortened electron transfer pathways and directional sweat transport^{32,33}.”

29. Qi, X. et al. Underwater sensing and warming E-textiles with reversible liquid metal electronics. *Chem. Eng. J.* **437**, 135382 (2022).

30. Li, G. et al. Autonomous Electroluminescent Textile for Visual Interaction and Environmental Warning. *Nano Lett.* **23**, 8436-8444, (2023).

31. Zhang, S. et al. Aid of Smart Nursing to Pressure Injury Prevention and Rehabilitation of Textile Cushions. *Adv. Fiber Mater.* **6**, 841-851 (2024).

32. Cai, S. et al. Air-permeable electrode for highly sensitive and noninvasive glucose monitoring enabled by graphene fiber fabrics. *Nano Energy* **93**, 106904 (2022).

33. Wei, X. et al. Wearable biosensor for sensitive detection of uric acid in artificial sweat enabled by a fiber structured sensing interface. *Nano Energy* **85**, 106031 (2021).

Lines 320-322: authors should provide more experimental detail concerning the validation of proposed sensors. The instrumentation and the analytical procedure should be added in the experimental section.

Response: Thank you for the constructive remark. We have supplemented the experimental detail concerning the validation of biosensors in the revised manuscript.

Revision:

(Page 27 in the revised manuscript)

“4.9 Off-Body Sweat Analysis

Post-exercise sweat samples were collected from the same skin area over which the hairband had been positioned for off-body data validation. For ion-sensing results validation, 2 μ L sweat was applied to another biosensor array, and the subsequent potential responses were monitored using an electrochemical workstation (CHI760e). Ionic concentrations were derived through linear regression analysis ($y = kx + b$)

utilizing standard calibration solutions, where y represents the measured potential and x denotes the logarithmic ion concentration. The pH values were verified using a pH meter (LICHEN PH-10), while temperature measurements were acquired using a thermometer (WDKL-EWQ-004).”

Minor comments:

Line 14: which are the biomarkers?

Response: Thank you for the remark. The biomarkers previously mentioned in the original manuscript have been replaced by the “pH, Na⁺, K⁺, and Ca²⁺” in the revised manuscript.

Line 18: “electron transmission” is misleading in the field of electrochemical sensor. It is related to TEM

Response: Thank you for your valuable remark. In the revised manuscript, the content related to “electron transmission” has been deleted to enhance the rigor of the research.

Introduction:

Line 35: blood gas analyzer and biochemical analyzer should be replaced with a little discussion concerning the gold standard techniques of blood analyzers.

Response: Many thanks for the comments. The related content about blood gas analyzer and biochemical analyzer have been removed in the revised manuscript. Furthermore, we have reorganized the introduction part and focused on sweat analysis.

Lines 49-51: references should be added

Response: Thank you for the remark. We have supplemented Ref. 27 to support the opinion in the revised manuscript.

27. Gao, F. et al. Wearable and flexible electrochemical sensors for sweat analysis: a review. *Microsyst. Nanoeng.* **9**, 1 (2023).

Lines 51-53: references should be added

Response: Thank you for the remark. We have supplemented Ref. 28 to support the opinion in the revised manuscript.

28. Tian, H. et al. Electrochemical sensing fibers for wearable health monitoring devices. *Biosens. Bioelectron.* **246**, 115890 (2024).

Lines 57-59: The technology transfer from PET and PDMS to textile fibers needs to be discussed.

Response: Many thanks for the comments. We have reconstructed the introduction and supplemented the technology of transfer from PET and PDMS to textile fibers in the revised manuscript.

Revision:

(Page 3-4 in the revised manuscript)

“Subsequent developments have also employed flexible polymeric substrates, such as polydimethylsiloxane (PDMS)²¹⁻²³ and polyimide (PI)^{24,25}, to enhance mechanical compliance with epidermal surfaces. However, the intrinsic hydrophobicity of polymer impedes interfacial fluid transport, thereby undermining continuous biosensing reliability. While microfluidic integration enhances biofluid collection, intricate fabrication and high cost remain unresolved issues. Additionally, poor breathability frequently induces skin irritation during prolonged wear, presenting critical barriers the long-term application of wearable biosensors.

Textiles with hierarchical fiber architectures spanning micro-to-nano scales provide distinct advantages for wearable integration, particularly in breathability, lightweight, and intrinsic compatibility with garment systems²⁶. While conventional textile biosensor fabrication predominantly employs drop-coating or printing techniques for functional layer deposition, these methods often exhibit structural inhomogeneity and performance variability²⁷. The deficiencies are attributable to weak interfacial binding energy between textile substrate and functional layer, resulting in progressive delamination and performance degradation²⁸. Such interfacial failures represent fundamental limitations of additive manufacturing strategies based on post-fabrication functionalization. In contrast, wet spinning is a monolithic yarn formation technique, whereby spinning dope is extruded into a coagulation bath and further utilized solvent-nonsolvent dual diffusion to induce phase separation²⁹⁻³¹.”

21. X. Mei, J. Yang, J. Liu, Y. Li, Wearable, nanofiber-based microfluidic systems with integrated electrochemical and colorimetric sensing arrays for multiplex sweat analysis.

- Chem. Eng. J.* **454**, 140248 (2023).
22. Y. Song et al., 3D-printed epifluidic electronic skin for machine learning–powered multimodal health surveillance. *Sci. Adv.* **9**, eadi6492 (2023).
23. F. Lorestani et al., A Highly Sensitive and Long-Term Stable Wearable Patch for Continuous Analysis of Biomarkers in Sweat. *Adv. Funct. Mater.* **33**, 2306117 (2023).
24. J. Niu et al., A Fully Elastic Wearable Electrochemical Sweat Detection System of Tree-Bionic Microfluidic Structure for Real-Time Monitoring. *Small* **20**, 2306769 (2024).
25. Y. Gai et al., A Self-Powered Wearable Sensor for Continuous Wireless Sweat Monitoring. *Small Methods* **6**, 2200653 (2022).
26. L. Possanzini et al., Textile sensors platform for the selective and simultaneous detection of chloride ion and pH in sweat. *Sci. Rep.* **10**, 17180 (2020).
27. F. Gao et al., Wearable and flexible electrochemical sensors for sweat analysis: a review. *Microsyst. Nanoeng.* **9**, 1 (2023).
28. Tian, H. et al. Electrochemical sensing fibers for wearable health monitoring devices. *Biosens. Bioelectron.* **246**, 115890 (2024).
29. Qi, X. et al. Underwater sensing and warming E-textiles with reversible liquid metal electronics. *Chem. Eng. J.* **437**, 135382 (2022).
30. Li, G. et al. Autonomous Electroluminescent Textile for Visual Interaction and Environmental Warning. *Nano Lett.* **23**, 8436-8444, (2023).
31. Zhang, S. et al. Aid of Smart Nursing to Pressure Injury Prevention and Rehabilitation of Textile Cushions. *Adv. Fiber Mater.* **6**, 841-851 (2024).

Line 62: references should be added

Response: Thank you for the remark. We have supplemented Ref. 32 and 33 to support the opinion in the revised manuscript.

32. Cai, S. et al. Air-permeable electrode for highly sensitive and noninvasive glucose monitoring enabled by graphene fiber fabrics. *Nano Energy* **93**, 106904 (2022).
33. Wei, X. et al. Wearable biosensor for sensitive detection of uric acid in artificial sweat enabled by a fiber structured sensing interface. *Nano Energy* **85**, 106031 (2021).

Lines 63-64: references should be added

Response: Thank you for the remark. We have supplemented Ref. 34 and 35 to support

the opinion in the revised manuscript.

34. Mo, L., Ma, X., Fan, L., Xin, J. H. & Yu, H. Weavable, large-scaled, rapid response, long-term stable electrochemical fabric sensor integrated into clothing for monitoring potassium ions in sweat. *Chem. Eng. J.* **454**, 140473 (2023).

35. He, X., Fan, C., Xu, T. & Zhang, X. Biospired Janus Silk E-Textiles with Wet–Thermal Comfort for Highly Efficient Biofluid Monitoring. *Nano Lett.* **21**, 8880-8887, (2021).

Lines 79-81: these data are not appropriate for introduction section

Response: Thanks for the suggestion. These data have been replaced with the sensitivities of pH, Na⁺, K⁺, Ca²⁺ and temperature biosensors in the revised manuscript.

Line 185: how can a scp solution be used as current collector?

Response: Thank you for the valuable remark. We sincerely apologize for the confusion caused by our unclear expressions in the manuscript. It has been determined that the SCP solution is to be corrected to the SCP mixture in the revised manuscript.

Line 204: the Ca⁺⁺ slope is 73.64. this value is higher than the Nernstian slope of 29 mV decade⁻¹. The author should provide a detailed explanation of it.

Response: Thank you for the constructive remark. We sincerely apologize for the negligent use of outdated data. The sensitivity of Ca²⁺ has been corrected to 30.61 mV dec⁻¹ after careful verification.

Line 297 figure 5: ambulance setting should be removed.

Response: Many thanks for the comments. The ambulance setting has been substituted for the photograph of the runner.

Revision:

(Page 22 in the revised manuscript)

Fig. 5c A subject wearing the hairband for health assessment.

Reviewer #3

Comments: The paper “Ultrasensitive and Stable Multi-Biosensing Hairband for Emergency Health Assessment” from M. Li et al reports a multi-analytes sensors realized using coaxial wet spinning technology, having silk yarn as the core. The as-spun sensors prove to have the target sensitivity range, flexibility, stability and capability of being integrated into a wearable hairband. The final configuration is also tested during exercise proving the achievement of real-time physiological analysis. Despite the work is interesting for the Nature Communication readers community and reaches a high technological level, the article still needs some investigations and several refinements before publication. The main suggestions and comments are reported here below:

Response: We thank the reviewer for the positive comments and valuable suggestions. Below, we respond individually to each specific point raised by the Reviewer and discuss how we have incorporated the reviewer’s suggestions into the revised manuscript.

1) Page 1, Abstract: The authors should specify the biomarkers and the range they monitor in the abstract.

Response: Thank you for the remark. We have specified the target biomarkers with their measurement ranges in the abstract to improve its quantitative clarity and informational value.

Revision:

(Page 1 in the revised manuscript)

“Blood analysis is regarded as the gold standard for monitoring analytes in clinical diagnostics. However, its time-consuming and site-limited nature often delays medical interventions that are crucial for patients in emergency scenarios. Herein, a novel multi-biosensor array is developed through coaxial wet spinning for real-time multiplex detection of sweat biomarkers (pH, Na⁺, K⁺, and Ca²⁺) and body temperature. The engineered microstructure of the multi-biosensor array exhibits a specific surface area approximately 200 times larger than that of conventional coated yarns, thereby facilitating directional sweat transport. The sensitivities of the pH, Na⁺, K⁺, Ca²⁺ and temperature sensors are determined to be 39.52 mV pH⁻¹ (3-7), 56.33 mV dec⁻¹ (10-160 mM), 34.13 mV dec⁻¹ (2-32 mM), 30.61 mV dec⁻¹ (0.5-2.53 mM), and 1.2 Ω °C⁻¹

(25-45 °C), respectively. It is noteworthy that the multi-biosensors exhibit consistent operational stability over 24 hours with minimal signal drift (pH:0.13 mV h⁻¹, Na⁺: 0.17 mV h⁻¹, K⁺:0.1 mV h⁻¹, Ca²⁺: 0.19 mV h⁻¹, temperature: 0.05 Ω h⁻¹). By integrating the multi-biosensors and a circuit patch into the textile substrate, a wireless hairband system is constructed for tracking human physiological dynamics. Such significant technological advancement offers an innovative strategy for constructing real-time biosensing systems, which have the potential to revolutionize personalized healthcare and early diagnosis, particularly in emergency situations.”

2) Page 4, "The drifts of the as-spun multimodal sensors are 1.87, 1.36, 2.3, 0.14 mV h⁻¹ for pH, Na⁺, K⁺, Ca²⁺ sensing, and 0.43 Ω h⁻¹ for temperature sensing,": Instead of focusing on the drift of the sensors, I would rather insert their sensitivity (eventually with the comparison of this drift of the baseline if needed).

Response: We thank the reviewer for this suggestion. The sensitivity of the as-spun multimodal sensors has been supplemented in the corresponding Introduction section, with the drifts of pH, Na⁺, K⁺, Ca²⁺ and temperature sensors subsequently repositioned in the revised manuscript

Revision:

(Page 5 in the revised manuscript)

“The multimodal biosensors, fabricated through coaxial wet spinning technology, achieve high sensitivities of 39.52 mV pH⁻¹ (pH), 56.33 mV dec⁻¹ (Na⁺), 34.13 mV dec⁻¹ (K⁺), 30.61 mV dec⁻¹ (Ca²⁺), and 1.2 Ω °C⁻¹ (temperature), respectively.”

3) Page 5, "Subsequently, as-spun multimodal sensor is knitted into a multiplexed sensing hairband system in parallel, thus effectively avoiding the short circuit caused by electrode contact": What do the authors mean by knitting in parallel to avoid short circuit? This description and fabrication process is not so clearly reported.

Response: Thank you for the valuable remark. The multi-biosensing hairband possessed a dense bilayer architecture based on the interlock stitch, which was woven by the SILVER REED SK280 sweater knitting machine (Fig. R1a). The interlock stitch comprised two rib stitches alternately arranged in opposite directions, with the inner layer consisting of interlaced sensing yarns and the outer layer functioning as a conventional textile substrate. It was evident that the longitudinal rows of one rib stitch

coil were arranged between the longitudinal rows of the other rib stitch coil. The interlacing pattern formed by the vertical warp and horizontal weft yarns created multiple interweaving points between the sensing yarns and the textile substrate, providing both structural reinforcement and positional stability (Fig. R1b). According to Munden’s geometric model, the quantitative relationship between the pitch and the diameter of the coil was as follows:

$$\frac{L}{W} = 0.72 \frac{W}{d} - 2 \frac{d}{W}$$

where L was the transverse pitch, W was the longitudinal pitch, and d was the yarn diameter.

The robust interfacial bonding force between these two layers ensured positional stability of the sensing yarns within the predetermined configuration. The adjacent yarns maintained a certain spacing under stretching, squeezing and bending, thereby effectively preventing contact-induced circuit failures (Fig. R1c).

With the reviewer’s suggestion, we have provided a detailed explanation of the anti-short-circuit principle of the hairband in the revised manuscript (highlighted in blue).

Fig. R1 Schematic illustration of the multi-biosensing hairband structure. a Continuous fabrication process of the biosensing fabric. Scale bar, 5 cm. b Schematic of the multi-

biosensing fabric structure and the knitted coil model. c Structural changes of the sensing fabric under extreme deformation.

Revision:

(Page 6 in the revised manuscript)

“Subsequently, these biosensors were knitted into a multiplexed sensing hairband system in parallel, which possessed a dense bilayer architecture (Supplementary Fig. 2). The interlacing pattern formed by the vertical warp and horizontal weft yarns created multiple interweaving points between the biosensing yarns and the textile substrate, providing both structural reinforcement and positional stability. The adjacent yarns can maintain a certain spacing even under extreme deformation, thus effectively avoiding the short circuit caused by electrode contact (Fig. 1b and c).”

4) Page 6, "the interfering ions, leading to the potential change at the total interface": The authors should better explain this mechanism. What is the potential change? What are the interfaces that are involved in this change and how it occurs?

Response: We thank the reviewer for this suggestion. The operation of potentiometric biosensors is typically achieved utilizing a dual-electrode system, which comprises a sensing electrode and a reference electrode. Following the capture of sweat, the ionophore-doped biosensor forms a loop system with the sweat, thereby converting the activity of the ions into an electrical potential as the output signal. The ionophore Nc , as the core part, possesses strong selective recognition ability for the target ion and will specifically absorb the target ion M^+ to form MNc^+ . Following complexation, MNc^+ reaches the contact interface and further transfers electrons to the conductive substrate. Depending on the types of ion-to-electron transduction, two different mechanisms exist: redox-reaction based and double-layer capacitance based transducers. The former involves redox reactions, while the latter provides a double electric layer at the contact interface. When the ion content reaches a certain level, a potential difference is generated at the overall interface.

We have added a detailed explanation in the revised manuscript (highlighted in blue).

Fig. R2 Sensing mechanism of potentiometric biosensors in sweat.

Revision:

(Page 7 in the revised manuscript)

“Upon sweat capture, the ionophore-doped biosensors formed a loop system with sweat, converting the activity of ions to potential as output signal. The ionophore demonstrated strong selectivity towards target ions, resisting interference from coexisting ions. It also facilitated charge transfer via electron migration, thereby leading to a measurable shift in the overall interfacial potential (Fig. 1e).”

5) Page 8, “SF/CNT/PLA (CSCP)”: The authors should add a cross-section figure to better explain the structure of the final core-sheath structured yarn.

Response: Thank you for the valuable remark. The cross-section SEM images of as-spun biosensors have been supplemented as shown in new Supplementary Fig. 7 below. The SCP-based biosensor was spun by extruding functional SCP mixture into coagulation bath, resulting in a homogeneous yarn architecture with controlled diameter. In contrast, the CSCP-based biosensor was fabricated through the coaxial wet spinning strategy. The custom-designed coaxial needle features two isolated microchannels: a central channel for continuous silk yarn feeding and a concentric annular channel for concurrent deposition of functional SCP sheath. Schematic diagrams and cross-sectional SEM images revealed that both the SCP and CSCP based biosensors exhibited exceptional structural stability, morphological consistency, and compositional homogeneity. Specifically, the CSCP-based biosensor possessed an optimized core-sheath structure with a central silk yarn core circumferentially sheathed by the functional SCP mixture. The structural design is attributed to the cohesive properties of the viscous spinning dope, thus providing superior resistance to structural failure or interfacial delamination, maintaining structural continuity during operational stresses.

We have added new Supplementary Fig. 7 in the revised supplementary information, along with a related description in the revised manuscript (highlighted in blue).

Supplementary Fig. 7 Schematic illustration and structural characterization of as-spun biosensors. a, b Schematic diagrams of SCP-based biosensor (a) and CSCP biosensor (b). c, d Cross-section SEM images of SCP-based biosensor (c) and CSCP biosensor (d). Scale bars, 100 μm .

Revision:

(Page 9 in the revised manuscript)

“Schematic diagrams and cross-sectional SEM images revealed that both SCP and CSCP based biosensors exhibited exceptional structural stability, morphological consistency, and compositional homogeneity. Specifically, the CSCP-based biosensor possessed an optimized core-sheath structure with a central silk yarn, which may provide a certain mechanical support (Supplementary Fig. 7).”

6) Page 8, "stress-strain curves": It looks like there are no differences among the yarns having different percentage of CNT. Is it the case? Can the authors comment on that and explain why the stretchability/flexibility does not depend on CNT concentrations?

Response: We thank the reviewer for this comment. The stress-strain curves of CSCP yarns demonstrated a significant non-monotonic dependence of breaking strength on carbon nanotube (CNT) concentration. At a concentration of 2 % CNT, the CSCP yarn exhibited a breaking strength of 34.33 MPa. With increased CNT concentration to 4-6 %, the breaking strength reached maximum values of 35.95 MPa and 35.79 MPa, respectively. The CNTs acted as a nano-reinforcement, thereby enhancing the strength of the yarns through an effective stress transfer mechanism. The mechanical performance deteriorated at 8 % CNT loading, with breaking strength declining to

34.53 MPa (4 % reduction from peak value). The decreased breaking strength may be attributed to the agglomeration of CNTs, which was induced by van der Waals forces. This agglomeration resulted in the formation of local stress concentration points, thereby disrupting the continuity of the yarns. The results demonstrated an initial strengthening phase (2-6 %), followed by property degradation at higher loadings (>6 %), establishing 4-6 % as the optimal CNT concentration. This concentration-dependent behavior aligns with previous research.

7) Page 8, "breaking strength of 35.95 MPa": It is not clear which is the curve the authors refer into the graph, since there are different curves having diverse percentages. What are those numbers referred to? If multiple tests have been done and some statistical analysis has been carried out for these breaking strength tests, the authors should provide the average values and the absolute error associated.

Response: Thank you for the constructive remark. We apologize for the oversight in the related descriptions of the tensile strength used in original manuscript. The mechanical characterization of breaking strength is conducted to validate the necessity of employing the core-sheath structure biosensor. Stress-strain analysis indicated breaking strength values of 34.33, 35.95, 35.79 and 34.53 MPa for CSCP yarns, compared to 1.66, 2.89, 2.11, and 1.81 MPa for SCP yarns. Consequently, the breaking strengths of these two architectures were quantified through mean values with standard deviation. The CSCP yarns demonstrated a mean breaking strength of 35.15 ± 0.73 MPa (n=4), representing a 16.6-fold enhancement over SCP yarns (2.12 ± 0.47 MPa, n=4). The considerable enhancement in mechanical strength substantiates the efficacy of coaxial spinning technology, positioning CSCP yarn as a superior candidate for textile-integrated biosensing applications.

Revision:

(Page 10 in the revised manuscript)

“Stress-strain analysis revealed that CSCP yarns exhibited significantly enhanced mechanical strength compared to SCP counterparts across varying CNT concentrations (Fig. 2e). Statistical quantification demonstrated CSCP yarns achieved 35.15 ± 0.73 MPa (n=4), representing a 17-fold enhancement over SCP yarns (2.12 ± 0.47 MPa, n=4). The considerable enhancement in mechanical strength substantiates the efficacy of the coaxial spinning technology, positioning CSCP yarn as a superior candidate for textile-

integrated biosensing applications.”

8) Page 12, "The sensitivities of the pH, ... 1.21 Ohm C-1": The authors should add the standard deviation of the sensitivities here reported. the Supporting Info should also report the complete fitting equation, with the intercept as well.

Response: Thank you for the valuable remark. The performance of as-spun biosensors is systematically evaluated through five repeated measurements. A quantitative analysis of sensing sensitivity was conducted using statistical methods, yielding the following values (mean \pm standard deviation, n=5): the pH sensor exhibited 39.52 ± 0.51 mV pH⁻¹, the Na⁺ sensor demonstrated 56.33 ± 1.23 mV dec⁻¹, the K⁺ sensor exhibited 34.13 ± 0.61 mV dec⁻¹, the Ca²⁺ sensor displayed 30.61 ± 0.82 mV dec⁻¹, and the temperature sensor presented 1.2 ± 0.02 Ω °C⁻¹. The calibration curves were established through linear fitting equation, which indicated distinct response characteristics. The pH sensor demonstrated an inverse linear relationship ($y=-39.53x+470.93$), while the Na⁺ and K⁺ sensors exhibited logarithmic dependencies expressed as $y=56.36\lg x+172.03$ and $y=34.17\lg x+219.88$, respectively. The Ca²⁺ sensor exhibited a proportional linear response of $y=30.71\lg x+251.07$, and the temperature sensor demonstrated a direct linear correlation with $y=1.2x+417.49$, with all fitting equations exhibiting excellent linearity ($R^2>0.99$, Supplementary Fig. 12).

With the reviewer’s suggestion, we have modified Supplementary Fig. 12 in the revised supplementary information, along with a related description in the revised manuscript (highlighted in blue).

Supplementary Fig. 12 Calibration curves of as-spun biosensor signals versus analyte concentration. a-e Linear response of the (a) pH, (b) Na⁺, (c) K⁺, (d) Ca²⁺ and (e) temperature sensors.

Revision:

(Page 13-14 in the revised manuscript)

“The sensitivities of as-spun pH, Na⁺, K⁺, Ca²⁺, and temperature biosensors were determined through five repeated measurements, yielding values of 39.52±0.51 mV pH⁻¹, 56.33±1.23 mV dec⁻¹, 34.13±0.61 mV dec⁻¹, 30.61±0.82 mV dec⁻¹, and 1.2±0.02 Ω °C⁻¹, respectively. Calibration curves for all as-spun biosensors were established via linear regression, demonstrating excellent linearity (R²>0.99, Supplementary Fig. 12).”

9) Page 12, "corresponding solution": What is the corresponding solution? Is it sweat, artificial sweat or simply a buffer solution?

Response: Thank you for the constructive remark. We have corrected the “corresponding solutions” for “standard solutions” in the revised manuscript.

10) Page 12, "reproducibility": The authors should calculate the reproducibility percentage or the reproducibility range, highlighting the deviation from the "standard"(average) behaviour that could be found in the sensors. Similarly, the authors should calculate the long-term stability for consecutive calibrations, better describing the normalized intensity that is reported in Supp Fig 9. What is this parameter and what is the amount of the decreased response?

Response: Thank you for the valuable remark. We have calculated the reproducibility percentage for highlighting the deviation from the standard behavior that could be found in the biosensors. The reproducibility of the sensitivity values of the pH, Na⁺, K⁺, Ca²⁺ and temperature sensors was found to be excellent, with a relative standard deviation (RSD) of 1.28%, 2.18%, 1.78%, 2.67%, and 1.87%, respectively (Supplementary Fig. 13). In addition, the long-term stability of the as-spun multimodal biosensors was calculated for consecutive calibrations, with measured performance degradation rates of 0.81% (pH), 1.43% (Na⁺), 1.15% (K⁺), 1.66% (Ca²⁺), and 0.95% (temperature) (Supplementary Fig. 16). Notably, the ion-selective sensors (Na⁺, K⁺,

Ca²⁺) exhibited relatively higher signal drifts compared to the others, likely induced by a reduction in ionophore activity.

With the reviewer's suggestion, we have added new Supplementary Fig. 13 and Fig. 16 in the revised supplementary information, along with a related description in the revised manuscript (highlighted in blue).

Supplementary Fig. 13 Reproducibility of as-spun biosensor signals versus analyte concentration. a-e Linear response of the (a) pH, (b) Na⁺, (c) K⁺, (d) Ca²⁺ and (e) temperature sensors.

Supplementary Fig. 16 Long-term stability of the as-spun biosensors within 60 days. a-e Change in sensitivity of the (a) pH, (b) Na⁺, (c) K⁺, (d) Ca²⁺ and (e) temperature sensors.

Revision:

(Page 14 in the revised manuscript)

“Furthermore, as-spun biosensors fabricated with the coaxial wet spinning process exhibited high reproducibility with relative standard deviations (RSD) of 1.28% (pH), 2.18% (Na⁺), 1.78% (K⁺), 2.67% (Ca²⁺), and 1.87% (temperature), as shown in Supplementary Fig. 13.”

(Page 14-15 in the revised manuscript)

“The long-term stability of as-spun multimodal biosensors was also confirmed through repeated measurements within 60 days, displaying signal degradation rates of 0.81% (pH), 1.43% (Na⁺), 1.15% (K⁺), 1.66% (Ca²⁺), and 0.95% (temperature) (Supplementary Fig. 16). It had been observed that ion-selective biosensors exhibited greater response drift, likely induced by a reduction in ionophore activity.”

11) Page 12, "previously reported patch or fiber": Why there is no comparison for the temperature sensor?

Response: Many thanks for the comment. We have completed the comparison between the as-spun temperature sensor and previously reported patch or textile biosensors (Supplementary Fig. 17). The sensitivity of the as-spun temperature sensor was 0.268% °C⁻¹, which surpassed the majority of existing wearable sensors (-1.04, 0.113, 0.204, 0.31, 0.05, and 0.214 % °C⁻¹) in the field.

With the reviewer’s suggestion, we have added new Supplementary Fig. 17 in the revised supplementary information, along with a related description in the revised manuscript (highlighted in blue).

Supplementary Fig. 17 Sensitivity comparison of as-spun temperature sensor with other reported patch or fiber sensors.

Revision:

(Page 15 in the revised manuscript)

“A comparison of the sensitivity between as-spun biosensors and previously reported patch^{22,24,25,41-53} or textile biosensors^{21,54-59} was presented in Fig. 3i and Supplementary Fig. 17.”

12) Page 13-15, Figure 4a: In Figure 4a the authors said the degradation analysis was done onto sensors: what is the sensor tested reported in the image? Are there differences among sensors into their degradation? Have the authors tested also their performances in terms of sensitivity and limit of detection? What are the white circles in the images? Is there a better image or higher resolution the authors can use? This looks very blurred.

Response: Thank you for the remark. The biodegradability of the as-spun biosensors was evaluated through soil-burial testing, with assessments conducted at 30-day intervals through optical characterization and weighing to quantify degradation progression. As shown in Figure R3, as-spun biosensors demonstrated time-dependent surface roughening and localized crack formation (marked with white circles), which was attributed to the synergistic effect of enzymatic hydrolysis of silk protein and ester bond cleavage of polylactic acid matrix. The use of a unified SCP substrate ensured consistent degradation kinetics, despite the functionalization through distinct sensing material doping. This hypothesis can be further verified by the comparison of mass loss rate (Fig.4b). To visually present the degradation process, Na⁺ sensing yarn was selected for an apparent comparative analysis at the single-yarn scale, clearly showing the evolution trajectory of the material from a complete structure to fragmentation.

In addressing the challenges associated with the recycling and treatment problem of waste sensing yarns, a fully biodegradable platform was engineered, leveraging the inherent degradability of silk protein and PLA. This type of yarn can gradually be converted into inorganic small molecules such as carbon dioxide and water during the natural degradation process, thereby effectively eliminating microplastic contamination risks in soil and marine ecosystems. Given that the measurement was specifically focused on the fundamental degradation characterization, the sensing performance was not monitored again during disintegration. The structure-property relationship governing sensing capability decay will be systematically investigated in subsequent research.

With the reviewer's suggestion, we have provided new Fig. 4a with high-resolution images, along with a related description in the revised manuscript (highlighted in blue).

Fig. R3 Optical images of as-spun biosensors subjected to degradation for 30 days.

Fig. 4a Optical images of the as-spun multimodal biosensors subjected to degradation.

Revision:

(Page 16 in the revised manuscript)

“As-spun biosensors demonstrated time-dependent surface roughening and localized crack formation, which was attributed to the synergistic effect of enzymatic hydrolysis of silk protein and ester bond cleavage of polylactic acid matrix (Fig. 4a).”

13) Page 14, "Following a 24-hour co-culture period": Why did the authors take the fluorescent images after 24h and not after 72h in order to have the parallel evaluation as the CCK-8 experiment? The authors should add these timings in the Figure caption.

Response: We thank the reviewer for this suggestion. We strongly agree that it is better to take the fluorescent images after 72 hours in order to enable a parallel evaluation with the CCK-8 experiment. Therefore, we have added the fluorescent images of mouse epithelial cells after 72 hours by co-culturing murine fibroblast L929 cells with the as-spun biosensors. As shown in new Fig. 4c below, live/dead staining assays revealed minimal cell mortality, indicating that the active materials integrated into the biosensors have no significant impact on cell viability.

With the reviewer's suggestion, we have added new Fig. 4c, along with a related description in the revised manuscript (highlighted in blue).

Fig. 4c Fluorescence microscopy images of mouse epithelial cells cultured in biocompatibility test (n=4 independent experiments). Scale bar, 100 μ m.

Revision:

(Page 16 in the revised manuscript)

“Following a 72-hour co-culture period with as-spun biosensors, fluorescent microscopy images were obtained.”

14) Page 14, Fig.4d: The authors should name the figure in the order they are cited, so this would be Fig. 4c.

Response: Thank you for the remark. We have corrected the figure numbers annotated for Fig. 4c and Fig. 4d in the revised manuscript.

Revision:

Fig. 4 Biocompatibility and robustness of as-spun multimodal biosensors. **a, b** Optical images (a) and weight loss rate (b) of as-spun multimodal biosensors subjected to degradation. **c, d** Fluorescence microscopy images (c) and CCK-8 experimental results (d) of mouse epithelial cells cultured for 72 hours (n=4). Scale bar, 100 μm . **e** Stability of as-spun multimodal biosensors under different deformations. **f** Optical image of as-spun biosensor embroidered silk handkerchief under gentle breeze. Scale bar, 5 cm. **g** Schematic of the multi-biosensing fabric structure. **h** Continuous fabrication process of the biosensing fabric. Scale bar, 5 cm. **i** Optical image of the tailored multi-biosensing hairband. Scale bar, 3 cm.

15) Page 18, "Following a 50-minute climbing session, sweat samples were collected to verify the accuracy": It is not clear if the comparison (for accuracy) has been done using the same as-spun sensors ex situ or a different as spun sensor or an analytical technique as validation. If no validation has been carried out, the authors should think of validating the sensors using a gold standard technique to assess these parameters.

Response: Thank you for the constructive remark. The accuracy of the biosensors has been validated in various ways. Ion-selective sensors were verified by another biosensor array using an electrochemical workstation (CHI760e), while pH values were verified using a pH meter (LICHEN PH-10) and temperature measurements were acquired using a thermometer (WDKL-EWQ-004).

With the reviewer's suggestion, we have supplemented the experimental detail concerning the validation of biosensors in the revised manuscript (highlighted in blue).

Revision:

(Page 27 in the revised manuscript)

“4.9 Off-Body Sweat Analysis

Post-exercise sweat samples were collected from the same skin area over which the hairband had been positioned for off-body data validation. For ion-sensing results validation, 2 μ L sweat was applied to another biosensor array, and the subsequent potential responses were monitored using an electrochemical workstation (CHI760e). Ionic concentrations were derived through linear regression analysis ($y = kx + b$) utilizing standard calibration solutions, where y represents the measured potential and x denotes the logarithmic ion concentration. The pH values were verified using a pH meter (LICHEN PH-10), while temperature measurements were acquired using a thermometer (WDKL-EWQ-004).”

16) Page 20, Conclusion: The authors should report some of the quantitative findings of the paper in the Conclusion for readers that do not read the whole manuscript but still need some detailed information, such for example the sensitivities of the sensors.

Response: Thank you for the remark. We have specified the sensitivities of as-spun biosensors in the Conclusion to enhance the quantitative clarity and informational value of the revised manuscript.

Revision:

(Page 23 in the revised manuscript)

“In summary, a weavable hairband integrated with as-spun multimodal biosensors for real-time physiological signal monitoring in emergency situations is demonstrated. The biosensors, fabricated through coaxial wet spinning technology, feature a seamless and uniform porous structure, significantly facilitating directional sweat transport and

electron transmission. Consequently, six signals can be monitored continuously and simultaneously. The sensitivities of pH, Na⁺, K⁺, Ca²⁺ and temperature biosensors have been determined to be 39.52 mV pH⁻¹, 56.33 mV dec⁻¹, 34.13 mV dec⁻¹, 30.61 mV dec⁻¹, and 1.2 Ω °C⁻¹, respectively. Furthermore, the minimal signal drift of 0.13, 0.17, 0.1, 0.19 mV h⁻¹ for pH, Na⁺, K⁺, Ca²⁺ sensing, and 0.05 Ω h⁻¹ for temperature sensing over 24 hours has been achieved. The utilization of eco-friendly raw materials and the incorporation of ergonomically designed structures ensure the favorable biodegradability and biocompatibility of the multi-biosensor array, enabling direct prolonged contact with human skin without causing irritation. As a proof-of-concept, the fabricated hairband system incorporates components for signal recording, processing, and wireless transmission, enabling real-time monitoring of human physiological conditions. This innovative design has the potential to be used for telemedicine monitoring and emergency health assessment, and would drive the innovation of the real-time health monitoring technology.”

Response to Reviewers for NCOMMS-25-01635

Reviewer #1

Comments: All my previous concerns were well addressed in a point-to-point way. The presentation of the results and the novelty is improved comparing with the origin version. Thus, I would like to recommend it for acceptance.

Response: We greatly thank the reviewer for the positive comment and the acknowledgment on our work. We also appreciate the constructive remarks and suggestions throughout the revision.

Reviewer #2

Comments: The manuscript can be published in the current form.

Response: We truly appreciate your offering this high praise for our paper. We are glad and humbled by the positive impression of our work, and thank you so much for your review.

Reviewer #3

Comments: The authors strongly improved the initial version of the manuscript and answered to all the Reviewers' comments. However, there are still some minor revisions I would suggest before publishing their results.

Response: We thank the reviewer for the positive comments and valuable suggestions. Below, we respond individually to each specific point raised by the Reviewer.

In particular, the minor concerns and adjustments are the following:

1) Abstract & Conclusion: The authors should report the sensitivities with the associated error; the same issue occurs at the end of the Introduction section. The same holds for the drift values that are reported without the errors.

Response: Many thanks for your helpful suggestion. We have checked the whole manuscript and supplemented the associated errors for the sensitivities and the drift values.

2) Errors should be reported using a single digit: this problem occurs throughout the whole manuscript.

Response: Thank you for the valuable remark. We have checked the whole manuscript

and revised errors with a single digit.

3)“Fig. R2 Sensing mechanism of potentiometric biosensors in sweat” Are the authors sure that the electrons derive into the silk fiber? My guess is that electrons are collected by the conductive carboxylated carbon nanotubes present in the SCP mixture, thus changing the electrode/material potential. This needs to be clearly reported in the main text as well.

Response: Thank you for the constructive suggestion. We sincerely apologize for the mistake of the inaccurate description of the sensing mechanism. The electrons are collected by the carboxylated carbon nanotubes present in the SCP mixture, thus changing the electrode/material potential. We have added a clear report of the sensing mechanism in the revised manuscript (highlighted in blue).

Revision:

(Page 7 in the revised manuscript)

“It also facilitated charge transfer via electron migration, enabling the charge collection by the conductive carboxylated carbon nanotubes (CNT) within the SF/CNT/PLA (SCP) mixture, which resulted in a measurable shift in the overall interfacial potential (Fig. 1e).”

4)“Standard solutions”: the authors should state and report what they mean for standard solution. Is it Phosphate Buffer Saline?

Response: We thank the reviewer for this comment. “Standard solutions” have been replaced by “phosphate buffer saline (PBS) solutions containing different analytes”.

5)“Electron transmission”: The authors should use “electron transport” instead, since it is misleading as electron transmission may refer to imaging techniques.

Response: Thank you for the constructive remark. “Electron transmission” has been replaced by “electron transport”.